# Remote biomass burning dominates southern West African air pollution during the monsoon

Sophie L. Haslett[1,a], Jonathan W. Taylor[1], Mathew Evans[2,3], Eleanor Morris[2], Bernhard Vogel[4], Alima Dajuma[4,5], Joel Brito[6], Anneke M. Batenburg[7], Stephan Borrmann[7], Johannes Schneider[7], Christiane Schulz[7], Cyrielle Denjean[8], Thierry Bourrianne[8], Peter Knippertz[4], Régis Dupuy[9], Alfons Schwarzenböck[9], Daniel Sauer[10], Cyrille Flamant[11], James Dorsey[1,12], Ian Crawford[1] and Hugh Coe[1].

[1] School of Earth and Environmental Sciences, University of Manchester, Manchester, United Kingdom
[2] Wolfson Atmospheric Chemistry Laboratories, Department of Chemistry, University of York, York, United Kingdom
[3] National Centre for Atmospheric Science, University of York, York, United Kingdom
[4] Institute of Meteorology and Climate Research, Karlsruhe Institute of Technology, Karlsruhe, Germany
[5] L'Université Félix Houphoët-Boigny, Abidjan 01, Côte D'Ivoire
[6] Laboratory for Meteorological Physics (LaMP), University Blaise Pascal, Aubière, France
[7] Particle Chemistry Department, Johannes Gutenberg University Mainz/Max Planck Institute for Chemistry, 55128 Mainz, Germany
[8] CNRM, Université de Toulouse, Météo-France, CNRS, Toulouse, France
[9] Laboratoire de Météorologie Physique, Université Clermont Auvergne, Aubière, France
[10] Institut für Physik der Atmosphäre, Deutsches Zentrum für Luft- und Raumfahrt, Oberpfaffenhofen Wessling, Germany
[11] LATMOS/IPSL, Sorbonne Université, UVSQ, CNRS, Paris, France
[12] National Centre for Atmospheric Science, University of Manchester, Manchester, United Kingdom
[a] now at: Department of Environmental Science and Analytical Chemistry, Stockholm University, Stockholm 11418, Sweden

*Correspondence to*: Hugh Coe (hugh.coe@manchester.ac.uk)

**Abstract.** Vast stretches of agricultural land in southern and central Africa are burnt between June and September each year, which releases large quantities of aerosol into the atmosphere. The resulting smoke plumes are carried west over the Atlantic Ocean at altitudes between 2 and 4 km. As only limited observational data in West Africa have existed until now, whether this pollution has an impact at lower altitudes has remained unclear. The Dynamics-Aerosol-Chemistry-Cloud Interactions in West Africa (DACCIWA) aircraft campaign took place in southern West Africa during June and July 2016, with the aim of observing gas and aerosol properties in the region in order to assess anthropogenic and other influences on the atmosphere.

Results presented here show that a significant mass of aged accumulation mode aerosol was present in the southern West African monsoon layer, over both the ocean and the continent. A median dry aerosol concentration of 6.2 µg m$^{-3}$ (standard temperature and pressure (STP)) was observed over the Atlantic Ocean upwind of the major cities, with an interquartile range from 5.3 to 8.0 µg m$^{-3}$. This concentration increased to a median of 11.1 µg m$^{-3}$ (8.6 to 15.7 µg m$^{-3}$) in the immediate outflow from cities. In the continental air mass away from the cities, the median aerosol loading was 7.5 µg m$^{-3}$ (5.9 to 10.5 µg m$^{-3}$). The accumulation mode aerosol population over land displayed similar chemical properties to the upstream population, which implies that upstream aerosol is a significant source of

aerosol pollution over the continent. The upstream aerosol is found to have most likely originated from central and southern African biomass burning. This demonstrates that biomass burning plumes are being advected northwards, after being entrained into the monsoon layer over the eastern tropical Atlantic Ocean. It is shown observationally for the first time that they contribute up to 80% to the regional aerosol loading in the monsoon layer over southern West Africa. Results from the COSMO-ART (COnsortium for Small-scale MOdelling – Aerosol and Reactive Trace

gases) and GEOS-Chem models support this conclusion, showing that observed aerosol concentrations over the northern Atlantic Ocean can only be reproduced when the contribution of transported biomass burning aerosol is taken into account.

As a result, the large and growing emissions from the coastal cities are overlaid on an already substantial aerosol background. Simulations using COSMO-ART show that cloud droplet number concentrations can increase by up to

27% as a result of transported biomass burning aerosol. On a regional scale this renders cloud properties and precipitation less sensitive to future increases in anthropogenic emissions. In addition, such high background loadings will lead to greater pollution exposure for the large and growing population in southern West Africa. These results emphasise the importance of including aerosol from across country borders in the development of air pollution policies and interventions in regions such as West Africa.

**1 Introduction**

West Africa is currently undergoing rapid urbanisation, population growth and industrial development. As a result of these large socioeconomic changes, anthropogenic pollution in the region tripled between 1950 and 2000 (Lamarque et al., 2010), and is expected to do so again from 2005 to 2030 (Liousse et al., 2014). Nevertheless, West African air quality is among the most poorly studied worldwide. As a result, these changes are being imposed on a largely

unknown regional background (Zuidema et al., 2016; Knippertz et al., 2015).

Plumes of biomass burning pollution from further afield are known to impact the mid-troposphere above West Africa during the summer monsoon (Chatfield et al., 1998; Mari et al., 2008, Murphy et al., 2010; Reeves et al., 2010; Sauvage et al., 2005). These plumes are the result of vast quantities of agricultural land in southern and central Africa being burnt between June and September each year (Barbosa et al., 1999). The large-scale burning releases

large quantities of aerosol into the atmosphere, which are carried west over the Atlantic Ocean at altitudes between 2 and 4 km. This transport mechanism is reliant on the southern hemispheric African Easterly Jet; when the jet is active, vast intrusions of biomass burning pollution can be transported across the Atlantic Ocean, in some cases as far west as South America (Mari et al., 2008). Intrusions into southern West Africa have been well documented from in-situ and satellite data. To date, this phenomenon has been thought to be confined predominantly to layers between

2 and 4 km in altitude (Barbosa et al., 1999; Capes et al., 2008; Chatfield et al., 1998; Mari et al., 2008).

Though recent modelling studies (Deroubaix et al., 2018; Menut et al., 2018) indicate that pollution may mix further down into the boundary layer, this has remained unconfirmed due to limited in-situ observations. Recent attempts to

quantify the extent to which smoke over the Atlantic Ocean entrains into the boundary layer using model simulations have shown that different models can provide very different results (Das et al., 2017; Peers et al., 2016).

Deroubaix et al. (2018) suggest that long-range transport of biomass burning aerosol could have contributed around 50% of $PM_{2.5}$ (aerosol smaller than 2.5 µm in diameter) mass in southern West Africa during the monsoon season observed by the AMMA campaign in 2006. These results make the collection and analysis of observational evidence on this matter particularly important, to confirm and quantify the presence of long-range transported aerosol in the boundary layer.

Both near-field and remote sources of pollution are likely to have an effect on cloud properties, radiative forcing and human health in southern West Africa. During the onset of the West African Monsoon, aerosol becomes entrained into newly-forming banks of monsoon clouds, so could have a resultant effect on rainfall patterns as well as the region's response to climate change. The emergence of megacities along the southern coast means that large numbers of people will be exposed to any atmospheric pollutants that exist in the region.

Airborne measurements made during the Dynamics-Aerosol-Chemistry-Cloud Interactions in West Africa (DACCIWA) campaign (Knippertz et al., 2015; Flamant et al., 2018) in June - July 2016 provided the opportunity to map aerosol properties in southern West Africa extensively. Here, observations from the three aircraft employed during the campaign are used to examine the relative contributions of local and transported pollution towards the aerosol loading in the regional monsoon layer (<1.9 km) in southern West Africa. The relative contributions of
regional urban emissions and aged biomass burning aerosol from central and southern Africa towards this background are assessed, using both observational evidence and simulations from the COSMO-ART (COnsortium for Small-scale MOdelling – Aerosol and Reactive Trace gases) and GEOS-Chem models. Biomass burning aerosol advected inland from remote sources is found to be the key driver of particulate pollution in the monsoon layer over southern West Africa away from large urban centres. The effect of this influx of long-range pollution on clouds
forming during the monsoon period is assessed using the COSMO-ART model.

**2 Method**

**2.1 Airborne observations**

The DACCIWA aircraft campaign took place during June and July 2016 and focused on the highly populated southern coastal region of West Africa. Science flying began on 29[th] June and concluded on 16[th] July 2016. Three
aircraft took part in the campaign: the German Deutsches Zentrum für Luft- und Raumfahrt (DLR) Falcon 20, the French Service des Avions Français Instrumentés pour la Recherche en Environnement (SAFIRE) ATR-42 and the British Antarctic Survey (BAS) Twin Otter. All three aircraft were based at the military airport in Lomé, Togo (6.16°N, 1.25°E), though the ATR-42 flew to the Aéroport Félix Houphouët-Boigny in Abidjan, Côte D'Ivoire, twice, on the 6th and 11th July. In total, 50 scientific flights were carried out, which comprised 155 hours of
scientific sorties. The DLR Falcon completed 12 scientific missions during the campaign, the ATR-42 completed 20 and the Twin Otter 18 (Flamant et al., 2018). The aircraft campaign took place after the monsoon onset; at this time,

the bulk of the monsoon rainfall is typically north over the Sahel, with limited precipitation in southern West Africa. The measurement period in 2016 was characterised by a northward-shifted intertropical discontinuity, which likely resulted in less wet deposition than usual. This period has been described as Phase 2 of the 2016 West African Monsoon (Knippertz et al., 2017).

Several flights were conducted between Lomé and the air above a ground station in Savè, Benin (8.03 ºN, 2.48 ºE), around 250 km to the north-east, which were used to build up statistics on background aerosol concentrations and cloud-aerosol interactions. Other flight patterns included city emission flights, which targeted city plumes and flights over the sea. Flight paths for all three aircraft are shown in Fig. 1.

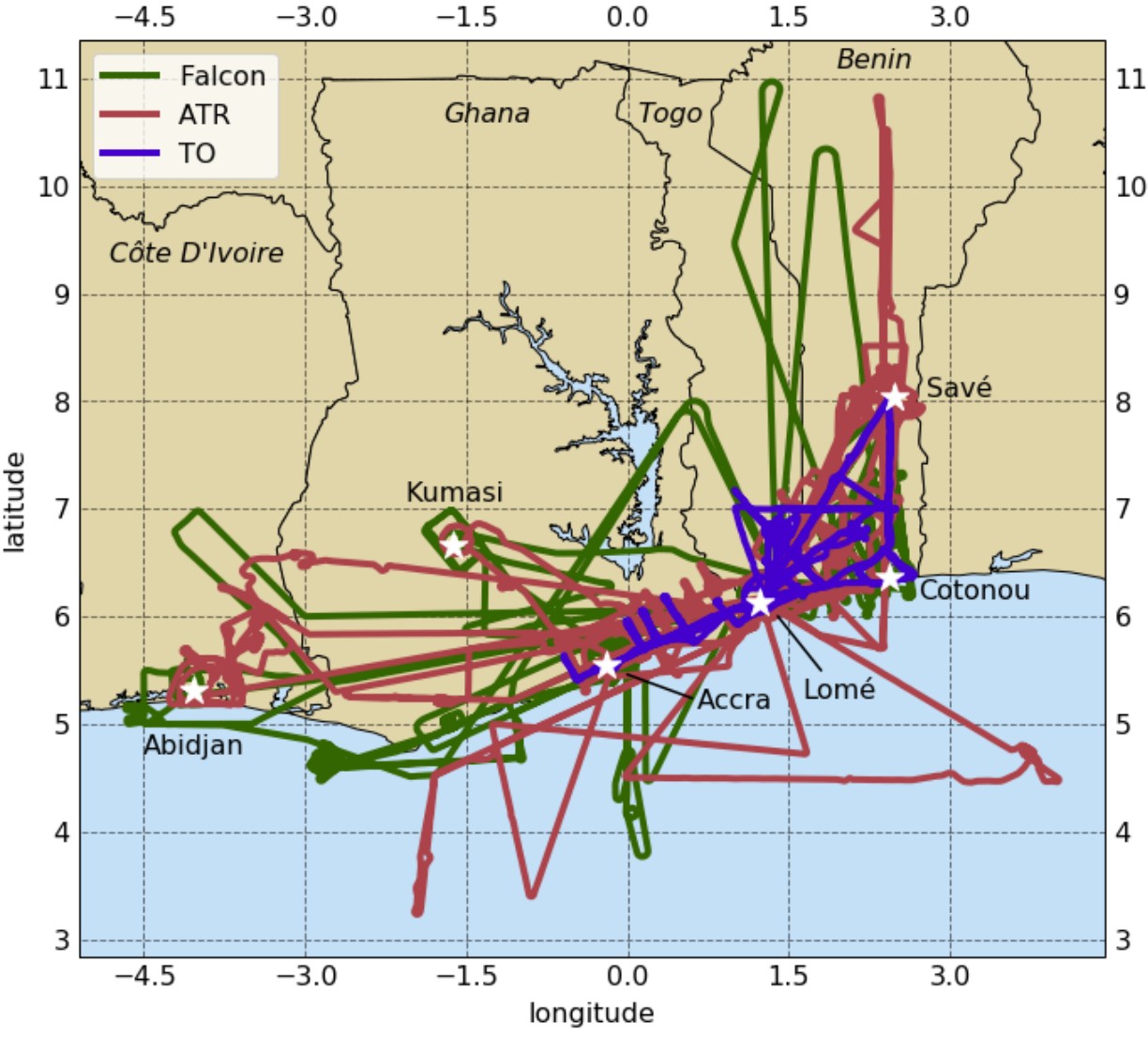

**Figure 1: Map showing the flight paths of the Falcon, ATR and Twin Otter aircraft during the DACCIWA campaign.**

Submicron aerosol chemical composition was measured using Aerodyne compact Time-of-Flight Aerosol Mass Spectrometers (AMS) (Drewnick et al., 2005; Canagaratna et al., 2007), mounted onboard all three aircraft. The instrument samples submicron particles from ambient air through an aerodynamic lens, which focuses the particles

in the vacuum chamber into a narrow beam. The particle beam is directed onto a hot surface vaporiser (~600 °C) where the particles are flash evaporated. The gas molecules formed are then ionised by electron ionisation and the ions are analysed in a time-of-flight mass spectrometer. The AMS produces quantitative chemical mass loading information for organic and inorganic non-refractory submicron aerosols with a time resolution of 20-45 seconds. A fragmentation table (Allan et al., 2004) was used to distinguish different compounds, yielding measurements of

sulfate ($SO_4$),  nitrate ($NO_3$), ammonium ($NH_4$) and organic compounds. An average collection efficiency of 0.5 was used throughout the campaign, which is standard for ambient measurements in similar environments (Middlebrook et al., 2012). The ionisation efficiencies of the instruments were calibrated several times througout the campaign using size-selected ammonium nitrate aerosol. A more detailed description of the ATR AMS data processing has been described by Brito et al. (2018).

Submicron aerosol size distributions were measured using a TSI Scanning Mobility Particle Sizer (SMPS) on board the ATR aircraft. This produced a size distribution of aerosol between 0.02 and 0.5 µm every 90 s. Condensation nuclei number concentrations were measured using Condensation Particle Counters (CPC) on board each of the three aircraft (a Brechtel Mixing CPC on the Twin Otter, modified TSI 3010 on the Falcon and TSI 3788 CPC on the ATR; lower size limits were 3 nm, 14 nm and 3 nm respectively).

In order to integrate these datasets successfully, the sensitivities of the instruments on all three platforms were compared. The transect between the coastal city of Lomé in Togo and the inland city of Savè in Benin was flown several times by each aircraft, which provided a basis for performing statistical comparisons. The median measurements of the AMS instruments on the ATR and the Twin Otter aircraft were within 20% of one another, although a larger interquartile range was observed in measurements from the ATR. The CPCs on board all three

instruments showed a discrepancy in the median values of less than 10%. Where applicable, measurements were corrected to standard temperature and pressure (STP).

The AMS data were compared for the take-offs and landings at Lomé airport. Despite calibration efforts, the AMS on the Falcon measured lower mass concentrations than the other two at low altitudes. This is believed to have been caused by a loss process at its inlet that affected the absolute, but not the relative measured mass concentrations of

the different compounds. Therefore, only the proportional chemical distribution and the high altitude mass concentrations from the Falcon AMS is used here. It is indicated in the text where these data are included.

The West African Monsoon governs surface level wind patterns in southern West Africa during June and July. Southerlies associated with the monsoon bring air into West Africa that has been advected over the ocean for several thousand kilometers (Williams et al., 2007). The incoming air is then affected by large coastal cities before

continuing inland. In order to study the influence of different sources, aerosol was analysed in three regimes:

'upwind marine', 'continental background' and 'urban outflow'. The first two include data collected above the ocean more than 20 km south of the shoreline and over West Africa away from immediate urban sources, respectively. This distinction provides a direct comparison between upwind air entering the region from the south and that influenced by the coastal cities, Abidjan (Côte D'Ivoire), Accra (Ghana), Lomé (Togo) or Cotonou (Benin).

The 'urban outflow' regime includes data from the centre of near-field urban plumes.

In all three cases, only aerosol below 1.9 km was considered. This is the height of the monsoon layer: the deep, moist layer that transports moisture into the continent from the south (Kalthoff et al., 2018). The boundary layer is the layer in direct exchange with the surface. This is typically shallow over the ocean (around 500 m); when the air reaches land, however, a much deeper mixing results in the low boundary layer over ocean mixing with the air

above it to deepen the boundary layer to around 1.5 km and to determine the concentration further inland. It is the monsoon layer below 1.9 km, however, that controls aerosol influx into the boundary layer over land, hence its use here. In the 'continental background' and 'urban outflow' regimes, data from below 100 m were removed to avoid bias, as the aircraft only flew at this altitude over land in the vicinity of the airport; the airport's influence was found to be negligible above this altitude. 'Urban outflow' data were from the centre of the near-field (<60 km) urban

plumes emitted from the cities listed above. These data were confined to include only measurements where $NO_x$ levels were within the highest 5% measured during the campaign (3.2 ppbv).

## 2.2   Regional Modelling

Two regional-scale models were used to test the hypothesis that aerosol measured by the aircraft over the sea is transported form central Africa: the COSMO-ART (Consortium for Small-scale Modelling – Aerosol Reactive

Trace gases) model is on online chemistry-transport model and is used to provide a high resolution (2.5km grid resolution) evaluation of a relatively short time period. This model has the advantage of being able to be used to investigate the impacts of biomass burning on cloud microphysical properties. GEOS-Chem (www.geos-chem.org) is an offline chemistry transport model run at a coarse resolution (0.25°x0.23125°), but with the advantage of being able to be run for a longer time period.

COSMO-ART is based on the German Weather Service (DWD)'s operational weather forecast model COSMO (Baldauf et al., 2011), coupled with an aerosol model (ART) for online treatment of aerosol chemistry and dynamics (Vogel et al., 2009; Bangert et al., 2012; Athanasopoulou et al., 2014; Knote et al., 2011). The interaction of aerosols with liquid and ice clouds were simulated using the two moment microphysics scheme of Seifert and Beheng (2006). For the liquid phase a parameterisation of Phillips et al. (2008) was applied (for details, see Bangert

et al. (2012) and Rieger et al. (2014)). This allows feedbacks between aerosols and radiation as well as between aerosols and clouds to be calculated.

Emission data from EDGAR (2010) (Emission Database for Global Atmospheric Research) were used for the anthropogenic emission of gases and aerosols. Natural emissions of biogenic volatile organic compounds (Weimer et al., 2017), sea salt (Lundgren et al., 2013), dimethyl sulfide (DMS; Lana et al., 2011), mineral dust (Stanelle et al.,

2010; Rieger et al., 2017) and GFAS (Global Fire Assimilation System) emissions from vegetation fires (Kaiser et al., 2012; Walter et al., 2016) are calculated online for each model time step. Gas flaring emissions are prescribed following Deetz and Vogel (2017). Meteorological initial and boundary conditions are taken from the operational global ICOsahedral Non-hydrostatic (ICON) model (Zängl et al., 2015) runs of DWD. Initial and boundary conditions for gaseous and particulate compounds are derived from Model for Ozone and Related Chemical Tracers

(MOZART) forecasts (Emmons et al., 2010).

There was a spin-up period of 7 days (19 June to 5 July 2016) and results are presented for 24 hours on 6 July 2016. Two simulations were performed for this study: one with the biomass burning emissions (both near-field and remote) included and the other without. The simulations were performed over a large domain (D1) covering West Africa and the south eastern Atlantic Ocean with a grid size of 5 km and 50 vertical layers. The output from D1 was

used to provide boundary conditions for a smaller, nested domain (D2) covering southern West Africa (the DACCIWA region), with a grid spacing of 2.5 km and 80 vertical levels (see Fig. 2). Although domain D1 covers a large area, COSMO-ART is still a limited area model. Comparing Fig. 2 with Fig. 8a it is therefore evident that D1 misses about 20% of the fire emissions of central Africa. However, this is likely compensated by the provision of boundary conditions for gaseous and particulate compounds of the global model system MOZART (2017).

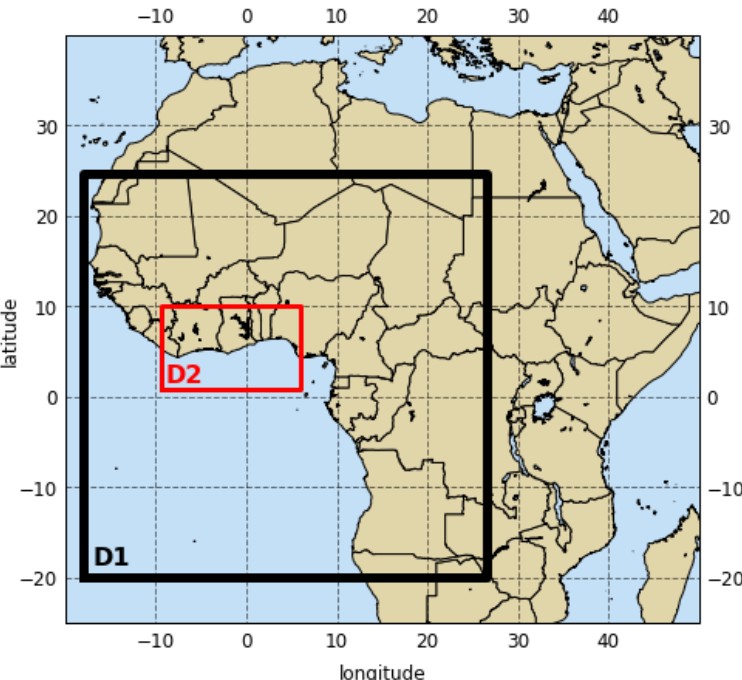


**Figure 2: The two nested domains used by the COSMO-ART model.**

GEOS-Chem is a three-dimensional model of tropospheric chemistry (Bey et al., 2001; Wang et al., 2004), driven with offline meteorological input from NASA Goddard Space Flight Center's Global Modeling and Assimilation Office. This study uses GEOS-Chem version 11-01 (http://wiki.seas.harvard.edu/geos-chem/index.php/GEOS-

Chem_v11-01). Simulations were performed globally at a horizontal resolution of 2° x 2.5° to provide boundary

conditions for the regional (latitudes 6°S - 6°N, longitudes 18.125°W – 26.875°E) West Africa model at a resolution of 0.25° x 0.3125°. Simulation have 47 vertical levels from the surface to 0.01 hPa, using meteorological data produced by the NASA Goddard Space Flight Center's Global Modeling and Assimilation Office. The model was run for the period from 29th June to the 16th July 2016 (the duration of the DACCIWA aircraft campaign) with a two week spin up period.

The model uses EDGAR v4.2 (EC-JRC / PBL, 2011) for anthropogenic emissions, which is overwritten by regional inventories where available: EMEP for Europe, NEI for the USA, CAC for Canada, MIX for South-East Asia and BRAVO for Mexico. Over Africa, the DACCIWA inventory (Junker & Liousse, 2008; Knippertz et al., 2015) is used for anthropogenic emissions. The Global Fire Assimilation System (GFAS) data is used for biomass burning emissions with a scale factor of 3.4 applied to the organic carbon emissions as recommended by Kaiser et al. (2012). The model uses MEGAN v2.1 (Guenther et al., 2012) for biogenic emissions of volatile organic compounds, biogenic soil $NO_x$ emissions from Hudman et al. (2012) and interactive lightning $NO_x$ (Murray et al., 2012). More details of the processing of organic aeorsol in the model can be found in Park et al. (2003).

### 3 Results and discussion

### 3.1  Observations

Figure 3 shows the aerosol particle number size distribution observed in each of the three regimes considered here. Significant variation can be seen in the number of smaller, Aitken mode particles, here considered to be those smaller than 100 nm. These particles are emitted from urban centres or formed from precursor gases and grow quickly in the atmosphere; large Aitken mode populations therefore indicate the presence of significant local sources. In the urban outflow regime, the Aitken mode concentration was often high, with a median number concentration of 3,400 $cm^{-3}$. In contrast, the Aitken mode was barely present in upwind marine air (median of 130 $cm^{-3}$). The number concentration of accumulation mode particles, with an average diameter near 200 nm, however, was remarkably consistent across the three regimes: 80% of the data lay within ± 30% of the campaign median in all cases. The median accumulation mode concentration was 600 $cm^{-3}$ STP in the upwind marine air and 850 $cm^{-3}$ STP in the continental background.

In the lower atmosphere during June and July, wind speeds in southern West Africa are generally low and wind comes from the south, becoming south-westerly as it approaches the coast. Therefore, within the monsoon layer, cool, moist Atlantic air progresses towards the cities and is likely to carry their plumes inland (Knippertz et al., 2017). Nevertheless, the similarity between the accumulation mode concentration in the upwind marine and continental background regimes seen here suggests that the air mass already contained a large number of the accumulation mode particles prior to urban influence. Comparing the median accumulation mode number concentrations in the continental background regime (850 $cm^{-3}$ STP) and the upstream marine regime (600 $cm^{-3}$ STP) suggests that far from the source, city emissions and land-based biogenic sources contributed only an extra 40% on top of the incoming accumulation mode aerosol. This calculation assumes a constant influence across the

region from incoming aerosol, and so likely represents a lower limit. Nevertheless, this implies that incoming pollution from the Atlantic has a considerable influence on the aerosol population over the land.

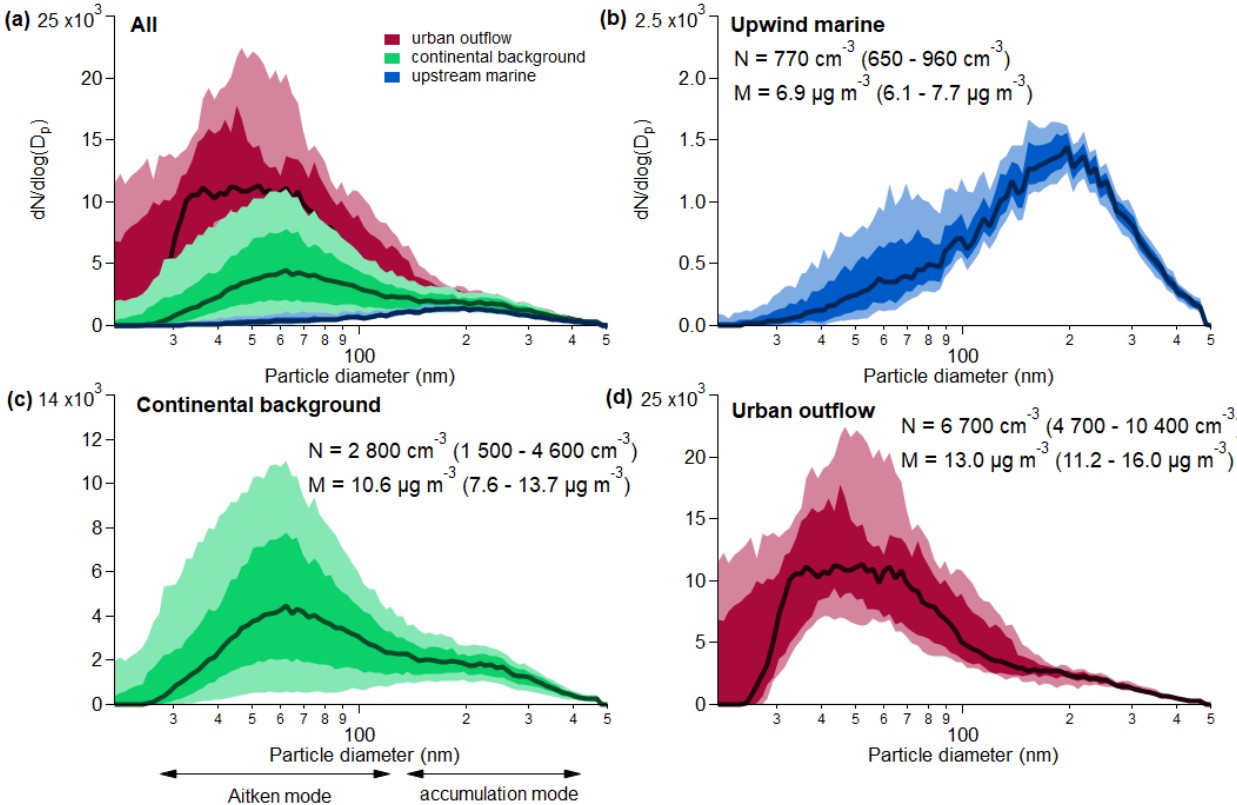

**Figure 3: Size distributions of aerosol in the urban outflow, continental background and upstream marine regimes, measured by the SMPS on board the ATR aircraft. For each regime, the median size distribution is shown by the dark line; the dark shading contains 50% of the data; and the light shading contains 80% of the data. The comparison of all three plots in panel (a) shows a stable accumulation mode that exists in all three regimes, centred at around 200 nm, while the smaller Aitken mode is much more variable. In panels (b-d), N shows the median total number concentration summed across the whole distribution, with the lower and upper quartiles shown in brackets; M shows the calculated aerosol mass, assuming an aerosol density of 1.6 g cm$^{-3}$ (Haslett et al., 2019), with the interquartile range again shown in brackets. The Aitken and accumulation modes are labelled in panel (c).**

The chemical composition of aerosols observed during DACCIWA supports the suggestion that much of the aerosol in the region originates upwind of the cities. Figure 4a shows aerosol mass and number concentrations in the three regimes, with mass classified by chemical species. The median total aerosol concentrations observed were 6.2, 11.1 and 7.5 µg m$^{-3}$ in the upwind marine, urban outflow and background continental regime, respectively, with interquartile ranges of 2.8, 7.1 and 4.2 µg m$^{-3}$. Although there was some day-to-day variability, which can be seen in the interquartile ranges shown in Fig. 3, there was no statistically significant variation in the median throughout the diurnal cycle (the variation in the median throughout the day was < 1 µg m$^{-3}$). Very little variation between the three regimes is seen in the proportional contribution of the different chemical species. The largest contribution to the measured PM$_1$ (particulate matter with a diameter smaller than 1 µm) was organic aerosol, which accounted for approximately 60% of the aerosol mass in all three regimes. Sulfate accounted for approximately 25% and nitrate

was generally low, comprising 4-6%. Ammonium contributed around 11%. The largest aerosol mass loadings and number concentrations were observed in the urban outflow, although even here, accumulation mode aerosol present in incoming air could account for as much as 50% of the total mass. Figure 4b shows the proportional chemical distribution for the three regimes explored here alongside those for the elevated biomass burning aerosol layer commonly sampled at 2-4 km from the ATR and Falcon, and the free troposphere above 5 km from the Falcon.

Figure 4c shows the modelled fine aerosol composition from COSMO-ART and GEOS-Chem. The COSMO-ART simulation is from 6 July, while that from GEOS-Chem is an average calculated from hourly output data from 29 June – 16 July 2016 (the duration of the DACCIWA aircraft campaign). The average concentrations in the upwind marine data (4.45 µg m$^{-3}$ for COSMO-ART and 5.47 µg m$^{-3}$ by GEOS-Chem) are in reasonable agreement with the observations (6.3 µg m$^{-3}$). In the urban outflow, modeled concentrations increase (11.52 µg m$^{-3}$ from COSMO-ART, 10.85 µg m$^{-3}$ from GEOS-Chem) and are again generally consistent with the observations (11.1 µg m$^{-3}$). In continental background air, however, there is a discrepancy, with COSMO-ART simulating 1.90 µg m$^{-3}$, while GEOS-Chem calculates 8.28 µg m$^{-3}$; observations found an average of 7.5 µg m$^{-3}$. Several factors could be responsible for the COSMO-ART response: (i) failure to simulate the inland progression of the marine air far enough northwards; (ii) overestimation of aerosol losses in the model, potentially due to vertical mixing being too strong over land; (iii) the model simulation being for a specific day, while observed results are from flights performed at different times during the DACCIWA campaign.

Switching off the biomass burning emissions over central Africa and West Africa reduces aerosol concentrations in both models. Models attribute ~75% of the upwind marine, and ~50% in the urban outflow of the aerosol mass to the biomass burning, with the vast majority of that occuring in central Africa (see Section 3.2). From these model studies we conclude that the majority of the fine aerosol seen in the upwind marine, and a significant fraction of that seen in the urban outflow is of biomass burning origin.

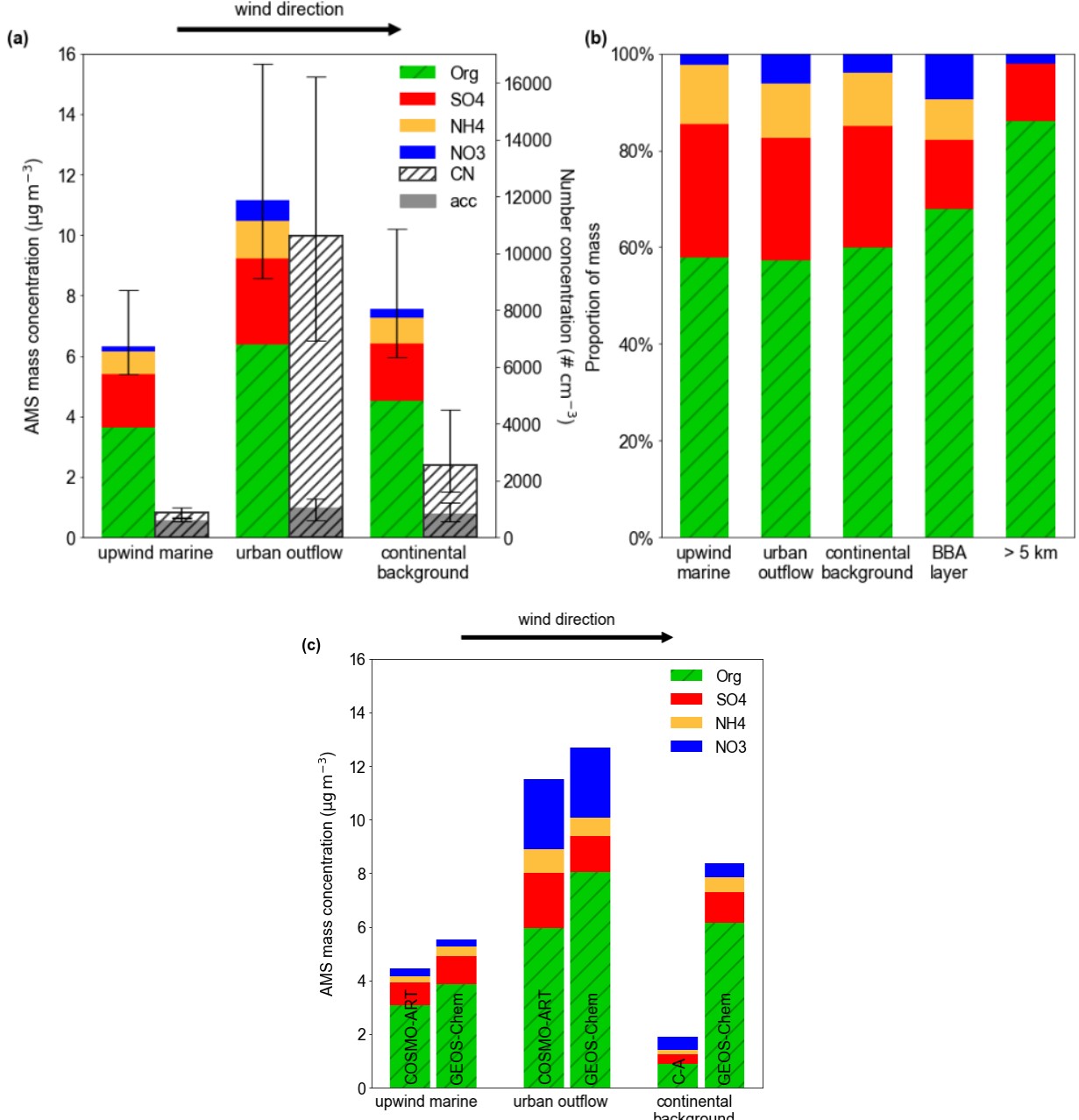

**Figure 4: (a) The observed chemical composition and condensation nucleus (CN) concentration in each of the three regimes. Coloured bars indicate aerosol mass concentration measured by the AMS. The CN bars show the total aerosol number concentration in each location (measured by the CPC), with shaded regions indicating the number of aerosol particles in the accumulation mode (derived from SMPS data shown in Fig. 3). Bars indicate the median, with error bars showing the interquartile range of observations (for total aerosol in the case of the AMS). A similar chemical distribution can be seen in each of the three regimes. (b) A comparison of the aerosol chemical distribution in all three regimes, alongside observations from the biomass burning layer at 3-4 km altitude (ATR and Falcon; labelled 'BBA layer') and the free troposphere (> 5 km; Falcon) for comparison. (c) Aerosol composition in the three regimes simulated by the COSMO-ART and GEOS-Chem models.**

The proportion of organic aerosol in the monsoon layer was large compared with what has been seen in other locations dominated by a mix of urban or biogenic emissions: Zhang et al. (2011) found that the global average

organic fraction measured by the AMS is between 43% in remote locations and 52% downwind from urban centres. The contribution of nitrate here, in contrast, was lower than is typically seen in locations influenced by urban outflow. Zhang et al. (2011) found a global average contribution of 12% and 18% in downwind and urban locations, respectively, which is much larger than was observed here. The high sulfate loading in the upwind marine regime is typical of marine aerosol. Sulfate over the oceans can originate from sea salt, as well as from the oxidation products of dimethyl sulfide (DMS) produced by phytoplankton. Non-sea-salt sulfate is typically found in concentrations of $0.2 – 1.5$ µg m$^{-3}$ in marine aerosol (Choi et al., 2017; Fitzgerald, 1991). In the inland regions, this likely mixes with sulfate produced from the cities.

The aerosol composition here is comparable with measurements during the dry season in South Africa, which can be influenced by emissions from savannah burning (Tiitta et al., 2014). The high contribution from organic aerosol is similar to observations of biomass burning plumes made in West Africa during 2006 as part of the DABEX (Dust and Biomass burning Experiment; Capes et al., 2008), as well as those made from biomass burning plumes in southern Africa (Vakkari et al., 2014). The key difference here is the large sulfate contribution, which is likely due to the influence of marine emissions.

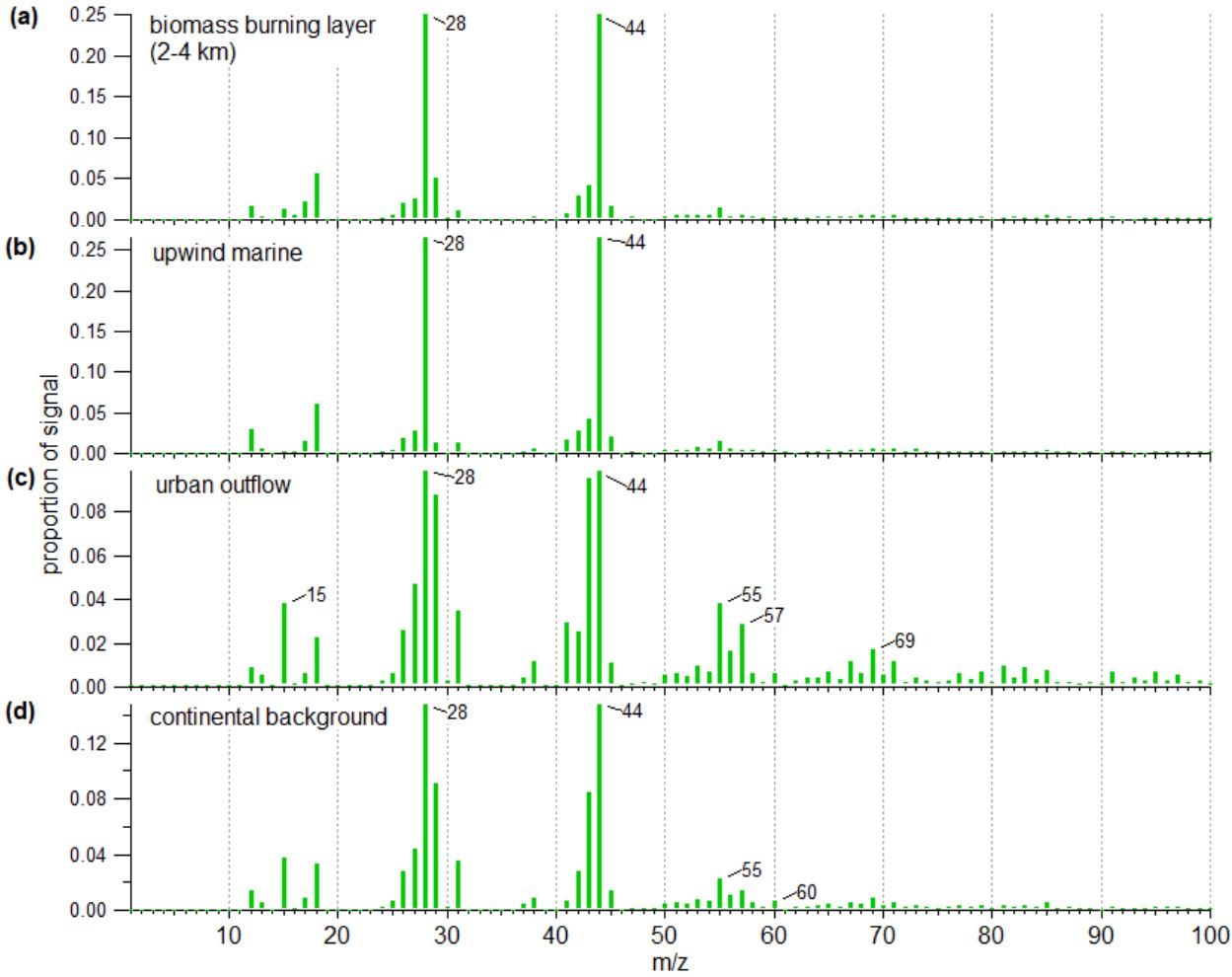


**Figure 5: Average organic aerosol mass spectra in (a) the 2-4 km biomass burning layer (from the Falcon AMS), and (b-d) each of the three regimes (ATR and Twin Otter). The influence of fresh urban emissions can be seen in the urban outflow regime and, to a lesser extent, in the continental background, demonstrated by the higher proportion of *m/z* 55 and 57, alongside larger hydrocarbon clusters. Fresh biomass burning is indicated by the peak at *m/z* 60. However, the**
**dominant contribution in all cases is from *m/z* 28 and 44, both indicators of aged, oxidised organic aerosol.**

The organic mass spectra from the AMS on board the Twin Otter and the ATR for the three regimes are shown in Fig. 5, alongside the mass spectrum of the biomass burning layer at 2-4 km from the Falcon AMS, which is widely considered to have originated from central Africa (Flamant et al., 2018). All four spectra are dominated by aged organic aerosol, which is characterised by strong peaks at *m/z* 28 and 44 (Ng et al., 2011). Although the 2-4 km

biomass burning layer showed a chemical distribution containing more organics and nitrate and less sulfate than aerosol observed lower in the atmosphere (Fig. 5b), the organic mass spectra shown here for this layer is very similar to that of the upstream marine aerosol, with *m/z* 28 and *m/z* 44 comprising 50% of the total organic mass in both cases. The urban outflow and, to a lesser extent, continental background mass spectra show features characteristic of urban pollution, including peaks at *m/z* 42, 55 and 91, which are associated with internal

combustion engines (Ng et al., 2011), and a number of clustered hydrocarbon peaks, for example at *m/z* 65, 67 and 69 or *m/z* 79, 81 and 83. A peak can be seen in the urban outflow and continental background regimes at *m/z* 60,

which is often associated with levoglucosan and other anhydrous sugars from biomass burning (Cubison et al., 2011) and likely arises from the widespread use of individual stoves for cooking, both in cities and in rural areas. This peak is associated only with fresh biomass burning; due to the oxidisation of anhydrous sugars in the

atmosphere (Henningan et al., 2011), it would no longer be strongly visible in the spectrum after a few days of processing (Cubsion et al., 2011). These features together indicate that local urban and fresh biomass burning sources do contribute to the aerosol mass loading in the monsoon layer over southern West Africa. However, there appears to be a further, significant, and considerably more aged source, which is entering the region from the south and has resulted in all four mass spectra being dominated by the oxidised peaks *m/z* 28 and *m/z* 44. This analysis is

supported by a PMF factor analysis that was carried out by Brito et al. (2018). The study identified a factor of highly-aged, oxidised organic aerosol, which was relatively homogeneously present across the entire DACCIWA campaign region, including over the Atlantic Ocean south of the coast.

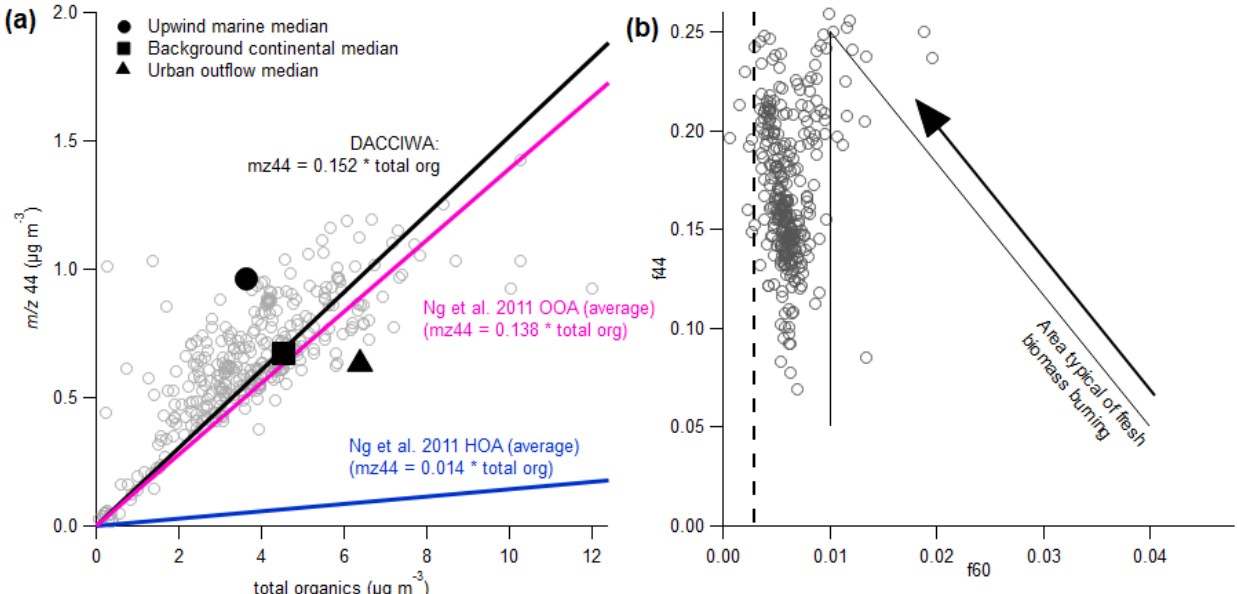

**Figure 6: (a) Values of *m/z* 44, a strong indicator of aged organic aerosol, plotted against the total organic aerosol mass.**
**The pink and blue lines show the average contribution of *m/z* 44 to the total for oxidised organic aerosol (OOA) and hydrocarbon-like organic aerosol (HOA). Grey circular markers show datapoints from the DACCIWA campaign. Median observations from each of the three regimes are shown as black shapes. (b) The fractional contribution of *m/z* 44 (f44) vs *m/z* 60 (f60). Fresh biomass burning aerosol is generally located within the triangle shown by the two black lines (Cubison et al., 2011) , with fresher aerosol lower in the triangle. The dashed line is representative of aerosol that is not**
**from fresh biomass burning. While *m/z* 60 aerosol is present here, the low contributions suggest that fresh biomass burning is not a dominant contributor towards the total organic aerosol mass.**

The dominance of aged aerosol in the overall population can be demonstrated further by comparing the magnitude of the *m/z* 44 peak with the total organic mass. Five-minute averaged datapoints from the Twin Otter aircraft are shown as markers in Fig. 6a. The dataset shown includes the continental background and urban outflow regimes; the

median observation for each regime, including upwind marine, is also displayed. The *m/z* 44 contribution here correlates well with the total organic mass (r = 0.78), with *m/z* 44 contributing around 15% throughout the campaign.

The relationship between $m/z$ 44 and the total organic aerosol mass can provide some insight into the source of ambient aerosol. The pink and blue lines in Fig. 6a show the averages across several campaigns from different parts of the globe for two different factors derived using positive matrix factorisation (PMF), as compiled by Ng et al. (2011). PMF is a technique used to analyse the contributions of different aerosol sources to an AMS dataset and identify a mass spectrum associated with each source based on its variation in time. Two factors commonly identified by PMF include oxidised organic aerosol (OOA) and hydrocarbon-like organic aerosol (HOA). The highly oxidised OOA factors are generally associated with photochemically aged aerosol, with $m/z$ 44 contributing a significant fraction of the total organic aerosol mass, as is shown by the higher gradient of the pink OOA line in Fig. 6a. The HOA fractions are often seen in pure fresh urban emissions. The contribution of $m/z$ to the total organic aerosol is significantly lower in these cases, with the $m/z$ 44 peak typically contributing less than 2% of the mass. This can be seen in the shallow gradient of the blue HOA line in Fig. 6a. In urban environments, mass spectra would be expected to have a large HOA component and thus, a large amount of scatter would be expected in the data, with the majority lying between the OOA and HOA lines. Here, the data are scattered predominantly around the OOA line, which suggests that the urban contribution is not the dominant factor in this dataset. The most significant proportion of the aerosol measured during the campaign is from aged, oxidised organic aerosol.

The presence of a peak at $m/z$ 60 in the continental background air suggests that local biomass burning is present in the observed air mass. This is explored in more detail in Fig. 6b. It has been shown previously that fresh biomass burning contains a large fraction of $m/z$ 60 ($f60$), a fragment of levoglucosan, which decreases as the plume ages. Furthermore, as the plume becomes more oxidised, the fraction of $m/z$ 44 ($f44$) increases. Thus, fresh biomass burning emissions populate the bottom of the triangle shown in Fig. 6b, and move towards the top corner in the direction shown by the arrow as they age (Cubison et al., 2011). The dashed line to the left shows the expected baseline values for air not containing any fresh biomass burning. Here, $f60$ is consistently slightly higher than the baseline, suggesting the presence of some fresh biomass burning. However, $f60$ is not high enough at any time to suggest that fresh biomass burning is the dominant source of aerosol. The values of $f44$ are generally high, which again shows the prevalence of aged organic aerosol in the air mass.

The relationship between the organic aerosol mass concentration and CO enhancement over the baseline ($\Delta CO$) is shown in Fig. 7a. Given its long atmospheric lifetime (Wang & Prinn, 1999), CO can be used as an inert tracer to account for the effects of dilution. A line of regression therefore indicates the source strengths and deviation from this line arises from changes in the organic aerosol due to photolytic effects, secondary organic aerosol (SOA) enhancements, wet removal or dry deposition. The intercept of the CO axis in this case within monsoon layer is at 0.11 ppmv, which is taken in further calculations to be the baseline across the region. The intercept above the monsoon layer was slightly lower, at 0.10.

The emission ratio $\Delta OA/\Delta CO$ (where $\Delta OA$ is organic aerosol enhancement above zero) in fresh urban plumes is usually lower than 20 µg m$^{-3}$ ppmv$^{-1}$, though once SOA has formed this increases to between 40 and 100 µg m$^{-3}$ ppmv$^{-1}$ (DeGouw and Jimenez, 2009). Unlike environments dominated by urban emissions, the $\Delta OA/\Delta CO$ in

biomass burning plumes is considerably more variable, with ratios having been observed between 45 μg m$^{-3}$ ppmv$^{-1}$ and 200 μg m$^{-3}$ ppmv$^{-1}$. Capes et al. (2008) reported an emission ratio of approximately 175 μg m$^{-3}$ ppmv$^{-1}$ in West

Africa during the DABEX campaign in 2006, while Vakkari et al. (2014) reported ratios between 50 and 200 μg m$^{-3}$ ppmv$^{-1}$ for biomass burning plumes in southern Africa. The high variability in ratios between different fires is likely related to the variable properties of the individual fire events (Jolleys et al., 2012), as well as to differences in atmospheric aging processes (Vakkari et al., 2014).

Here, the low emission ratio in the urban outflow (51 μg m$^{-3}$ ppmv$^{-1}$) is slightly higher than would be expected for a

fresh urban plume; this may be related both to the large amounts of biomass burning in the city and to the background aerosol. The ratio in the elevated biomass burning layer was considerably higher (225 μg m$^{-3}$ ppmv$^{-1}$). In the upwind marine (142 μg m$^{-3}$ ppmv$^{-1}$) and continental background (116 μg m$^{-3}$ ppmv$^{-1}$) regimes, the value was higher than would be expected from even an aged urban plume. Many of the datapoints in Fig. 7a fall along the same line as the elevated biomass burning layer. These results provide a strong sugestion that much of the aerosol

measured during the DACCIWA aircraft campaign did not originate from urban pollution, but from biomass burning.

Fig. 7b show modelled concentrations of CO from the GEOS-Chem model in the upwind marine regime, with the emissions from biomass burning switched off in the first model run and on in the second. This is shown alongside observed values of CO for each of the three regimes. The model results were only comparable with observations

when biomass burning emissions were included, suggesting a significant influence from biomass burning over the ocean.

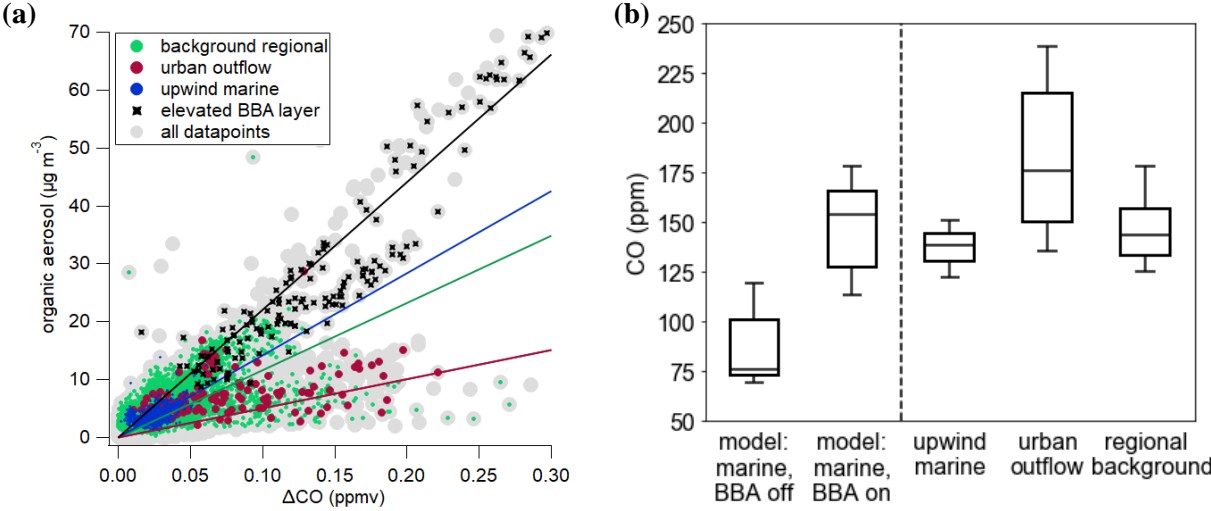

**Figure 7: (a) The ratio of organic aerosol to CO, with lines of regression included for each of the three regimes and the elevated BBA layer between 2 and 4 km. CO is shown as enhancement above the background (0.11/0.10 ppmv for**

**in/above the monsoon layer). (b) CO concentrations. Modelled results from GEOS-Chem show the CO concentration in the upwind marine regime with biomass burning emissions off/on. Observational results are shown from all three regimes. The model simulation is only comparable with observational measurements when biomass burning is considered.**

### 3.2 Southern and central African biomass burning

The lack of variability in the accumulation mode concentration and composition is evidence that much of the aerosol observed during the DACCIWA campaign originated from a similar type of source. Furthermore, the similarity between these characteristics across the three regimes, including the upwind marine, identifies the dominant source to be outside the urban coastal region, upwind of all three locations. A closer inspection of the organic aerosol mass spectra during DACCIWA suggests the presence of a large mass of aged aerosol, with smaller contributions from fresh urban and fresh biomass burning sources.

Aerosol in the upwind marine regime is unlikely to have originated from either the coastal cities or from oil fields over the Atlantic Ocean. The southerlies in the monsoon layer in this region are very stable, with an onshore flow during both the day and the night (Flamant et al., 2018; Guedje et al., 2019). Transport from land to sea is therefore largely impeded by the superposition of the sea breeze circulation with this strong southerly monsoon flow. As a result, city pollution is mostly transported inland and hardly reaches beyond 50 km south of the shoreline; even less when the southerlies are stronger (Flamant et al., 2018). A pilot balloon climatology by Guedje et al. (2019) for Cotonou reveals that during July-September the flow below 1 km is exclusively from the southerly quadrants, while at higher altitudes a weak northerly component occurs only occasionally. Radiosonde measurements from coastal stations show that there is hardly a northerly component in the wind direction at all (Flamant et al., 2018). Aerosol mass in the upwind marine regime is therefore unlikely to have originated from the cities and been transported south of the coast. Studies of oil rig emissions over the Atlantic carried out during the DACCIWA campaign show that these emissions are characterised by narrow plumes of pollution that are strongest close to the source, and which have generally dispersed after 40 km (Brocchi et al., 2019). This profile of spikes in CO, $NO_x$ and aerosol emissions was not observed during the flights that are, making it unlikely that they were influenced by oil field emissions. This evidence therefore shows that a large proportion of the aerosol mass in the continental West African boundary layer originates from the monsoon layer over the eastern tropical Atlantic Ocean, and is present prior to influence from coastal cities.

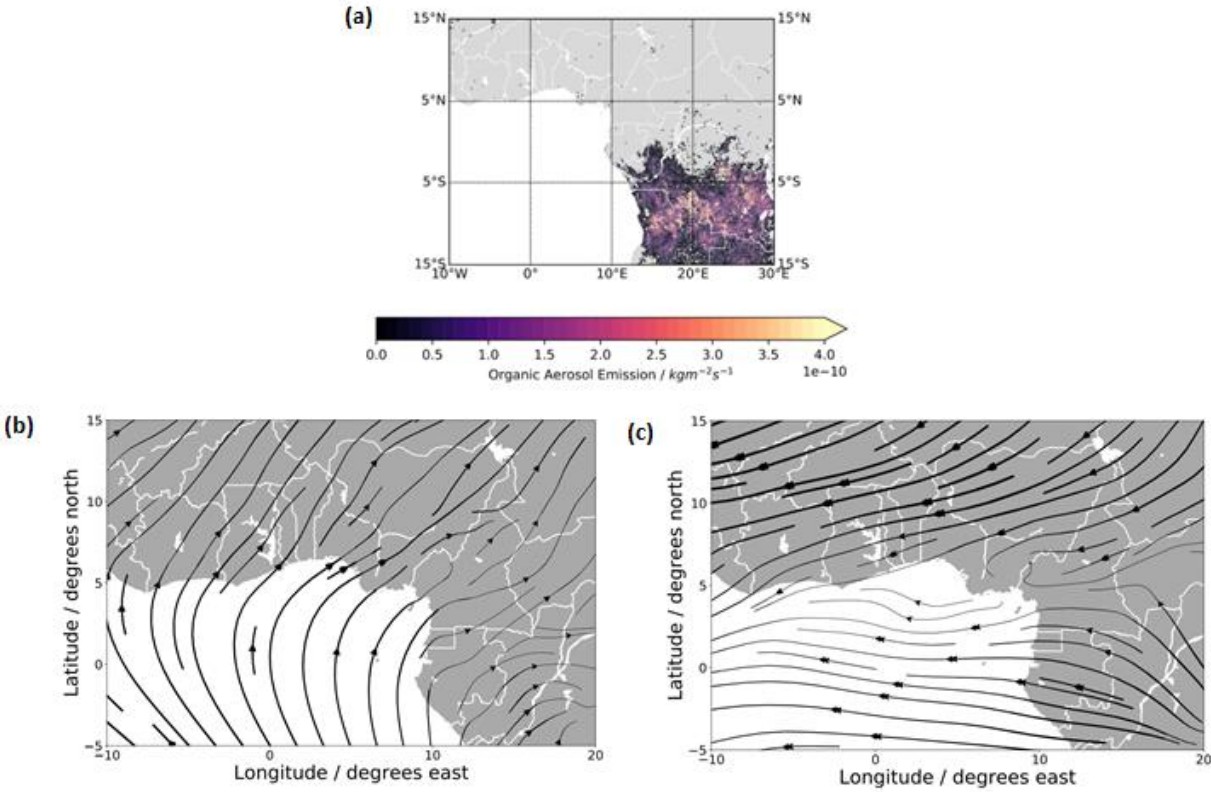

**Figure 8: (a) The location and intensity of organic aerosol emissions from biomass burning in central Africa during the DACCIWA campaign (June-July 2016) from the GFAS inventory (Kaiser et al., 2012), (b, c) Wind stream functions at (b) the surface level, and (c) 750 hPa (approx. 2.5 km) from the NASA Global Modelling and Data Assimilation Office's GEOS-5 analysis. The line thickness indicates wind speed.**

One of the most significant fine-mode aerosol sources south of the coastal cities is the agricultural and savannah burning that takes place annually in central and southern Africa between June and September (Barbosa et al., 1999). Vast quantities of biomass burning aerosol are injected into the mid troposphere between 3 and 4.5 km as a result of these fires (Labonne et al., 2007) and carried west over the Atlantic Ocean by tropospheric winds at around 700 hPa (Das et al., 2017; Edwards et al., 2006). The location and intensity of these fires during the DACCIWA aircraft campaign is shown in Fig. 8, alongside modelled wind streams at ground level and at 750 hPa. At 750 hPa, the easterly currents that carry biomass burning pollution west out of central Africa can be clearly seen, while the surface level chart shows the southerly air stream that passes from the Atlantic Ocean into the southern West African monsoon layer.

The observations of the aerosol composition in the continental monsoon layer during the DACCIWA campaign described above show it to be characteristic of aged biomass burning aerosol. The proportion of organic aerosol was higher and nitrate was lower than would be expected in areas influenced primarily by urban outflow (Zhang et al., 2011). The mass spectra of organic aerosol below 1.9 km were dominated by peaks typical of aged, low-volatility aerosol, which closely resembled the mass spectrum of the 2-4 km biomass burning layer. Even in the urban outflow

regime, mass spectral features associated with near-field urban sources such as internal combustion engines were less prominent than would be expected from pure urban aerosol. This evidence supports the assertion that these central and southern African fires are the primary source of accumulation mode aerosol in southern West Africa during the summer monsoon season.

Recent observations carried out on Ascension Island to the south-west of the DACCIWA region (7.93 °S, 14.42 °W) as part of the US Department of Energy Atmospheric Radiation Measurement Layered Atlantic Smoke Interactions with Clouds (LASIC) campaign show that smoke from these fires can be detected at the planetary surface (Zuidema et al., 2018). This demonstrates that the central and southern African biomass burning aerosol plume is commonly entrained into the monsoon layer of the remote tropical Atlantic Ocean to the south of the DACCIWA region. This

confirms that there is a pathway for biomass burning aerosol to enter the monsoon layer across large parts of the tropical eastern Atlantic. Once this biomass burning aerosol has been entrained into the monsoon layer, the prevailing southerly trade winds at the surface will carry it northwards towards the coast of southern West Africa. There was little evidence of precipitation over the east Atlantic and dry deposition rates of accumulation mode aerosol over open ocean are low. Once entrained, any aged biomass burning aerosol from central and southern

Africa would therefore be advected into the DACCIWA region with little further loss. It has been shown that biomass burning emissions in Africa are among the least variable in the world on annual timescales (Voulgarakis et al., 2015). This implies that this influence on the southern West African monsoon layer is likely to be a consistent feature of the West African Monsoon.

In a recent multi-model evaluation, the extent of the plume's entrainment into the monsoon layer over the Atlantic

Ocean proved difficult to model consistently (Das et al., 2017), with many models showing the plume descending too rapidly. In contrast, Gordon et al. (2018) use the HadGEM climate model to show the plume remaining above the clouds between 2-4 km and not descending at all until approximately 10 °W. A modelling study by Deroubaix et al. (2018) compared the impacts of long-range transported biomass burning aerosol with that of local anthropogenic pollution during the monsoon season observed by the AMMA campaign in 2006. In this study, it was found that

long-range transport of biomass burning aerosol was likely to contribute around 52% of the aerosol below 1 km in southern West Africa.

Results from the DACCIWA campaign verify the presence of regular biomass burning plume intrusions at altitudes of 2-4 km over the West African continent, with high aerosol loadings above 60 µg m$^{-3}$ being observed at this altitude in some cases (Flamant et al., 2018). This is consistent with research suggesting that the majority of the

southern and central African biomass burning plume remains above the clouds over the Atlantic Ocean (Adebiyi et al., 2015; Das et al., 2017; Gordon et al., 2018). However, results presented here show that in addition, a significant proportion of the aerosol mass from the biomass burning plume is being entrained into the monsoon layer, where it is likely to have a significant impact on cloud properties and human health, particularly for the large population living along West Africa's southern coast.

**3.3 Aerosol influence on cloud properties**

Clouds' susceptibility to increased accumulation mode aerosol decreases when an aerosol background already exists. The relationship between aerosol concentration and cloud droplet number concentration is governed by a power law (Duong et al., 2011; McComiskey et al., 2008; Ramanathan et al., 2001; Terai et al., 2012), so increasing the aerosol number concentration has a proportionally greater impact on clouds that would otherwise have formed in

clean air. Furthermore, the change in albedo from increasing the number of water droplets is greater in a cloud with an initially low concentration (Twomey, 1977). Below around 100 CCN cm$^{-3}$, light scattering by low-level cloud is extremely sensitive to even small increases in aerosol concentration (Kaufman and Fraser, 1997; Kaufman et al., 2005; Ramanathan et al., 2001). This susceptibility decreases gradually; a similar change from a higher initial loading will have a substantially smaller impact. Lower susceptibility would be expected for a cloud forming in a

region with an aerosol concentration of 600 particles cm$^{-3}$ or more.

During June and July, extensive low-level cloud forms along the West African southern coast (Knippertz et al., 2011; Schrage and Fink, 2012). It has been speculated that during the monsoon season, clouds above West Africa could be highly susceptible to increases in anthropogenic pollution (Knippertz et al., 2015). However, the presence of a significant quantity of biomass burning smoke in incoming wind from the Atlantic Ocean is likely to reduce this

effect.

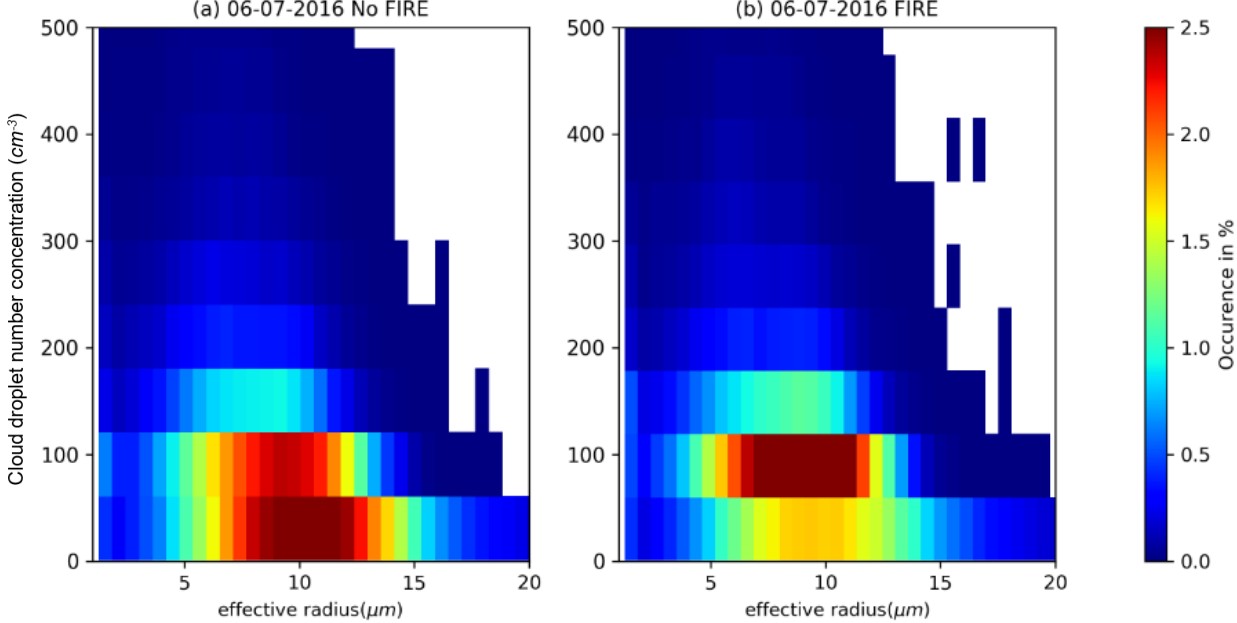

**Figure 9: Two dimensional histograms showing cloud properties in the NO FIRE (a) and FIRE (b) simulations conducted with COSMO-ART.**

Here, the effect of such an influx of biomass burning aerosol on cloud formation in the region was investigated

using the COSMO-ART model. Two simulations were carried out for 6 July: one including biomass burning aerosol

(FIRE) and one without (NO FIRE). These simulations are used here to illustrate the difference made to cloud properties by increasing the accumulation mode aerosol concentration. Figures 9a and b show two-dimensional histograms of cloud droplet number concentration and effective radius across the inner domain. In the NO FIRE case, the number concentrations are lower and the effective radii higher than in the FIRE case, with number concentrations increasing by up to 27%.


These results show that remote biomass burning aerosol creates a significant background loading that systematically perturbs the cloud field. Any increase in anthropogenic emissions will be superimposed onto this existing background, reducing its influence on cloud. Thus, while enhancements in cloud droplet number concentrations in near-field city plumes were observed during the DACCIWA campaign, the influence of these plumes became indistinguishable further afield as they dispersed into the background. On a regional scale, city plumes were of secondary importance. This conclusion is supported by observations of cloud droplet number concentrations carried out during the DACCIWA campaign (Taylor et al., 2019).


**4 Summary and conclusion**

Observations of aerosol measurements below 1.9 km were collated from the three aircraft that took part in the DACCIWA aircraft campaign during June and July 2016. A regional background of pollution was observed across southern West Africa, which contained around 6.3 µg m$^{-3}$ of dry aerosol in the accumulation mode and was dominated by aged organic matter. The lower atmosphere above the eastern tropical Atlantic Ocean, immediately upwind of the DACCIWA region, was similarly polluted. Mass concentrations of upwind pollutants here were typically around 80% of those over the land. Contributions from cities and local, small-scale biomass burning to the regional background was of secondary importance compared with this large aged aerosol mass. This aerosol background was attributed to large-scale biomass burning taking place in central and southern Africa. Emissions become entrained into the monsoon flow over the Atlantic Ocean and are advected northwards into the southern West African region. Aerosol concentrations simulated using the COSMO-ART and the GEOS-Chem models support this conclusion, showing that concentrations in the upwind marine and urban outflow regimes could not be replicated without remote biomass burning emissions being taken into account.




The chemical composition of this aerosol background is consistent with aged biomass burning being advected over the continent in the monsoon layer. Markers of oxidised aerosol dominated the organic mass spectra in all locations with a ratio to total organics that is typical for more aged aerosol. Urban aerosol and the signature of local biomass burning are present, but both play a minor role compared with the larger quantity of aged aerosol. Although there was some day-to-day variability in the total mass concentration, the aerosol background was observed across the entire region with very little variation in chemical composition, suggesting a large-scale, distant source. If this were related to locally-produced aerosol, greater variability would be expected across the region, with larger distinctions between urban outflow and rural measurements. Locally-produced aerosol would be unlikely to be observed over the ocean as far upwind of the coast as it has been observed here, and the composition of the upwind aerosol does


not resemble recycled urban emissions. It has been shown in previous DACCIWA studies that circulation of urban
       emissions over the ocean does not extend more than 50 km south of the southern West African coast (Flamant et al.,
       2018). Biomass burning from central and southern Africa is the most likely source of a large-scale mass of
       homogeneous aerosol in this region. This conclusion is consistent with observations from other campaigns that show
       biomass burning smoke is present at this time of year in the monsoon layer further south in Ascension Island
(Zuidema et al., 2018).

       Results presented here suggest that the biomass burning pollution accounts for up to 80% of the accumulation mode
       aerosol mass over the continent. Given this large moderating effect on the air pollution over West Africa at this time
       of year, the microphysics of the prevalent stratiform clouds in the West African Monsoon is likely already largely
       perturbed even before near-field anthropogenic pollution is taken into consideration. Simulations using the
COSMO-ART model showed significant differences in the cloud droplet number concentration and effective radius
       of cloud droplets when this biomass burning influx was taken into account. The cloud droplet number concentration
       increased by up to 27% over the marine domain when biomass burning was switched on.  This suggests that,
       significant increases in anthropogenic pollutants could have a smaller perturbing effect than would have been the
       case if incoming air were less polluted (Taylor et al., 2019).

This study takes place in the context of a strong focus in the research community on the dynamics and effects of the
       African biomass burning plume. A number of campaigns, including LASIC, ORACLES, CLARIFY and AEROCLO
       (Zuidema, 2016) have recently been carried out over the Atlantic Ocean west of the African continent, with the aim
       of better understanding this problem and quantifying the direct and semi-direct aerosol effects of the plume, which
       can differ significantly depending on the altitude at which the plume spreads (Das et al., 2017). This study provides
further motivation for understanding these processes, as it shows that the potential for biomass burning aerosol to
       become entrained into the southern West African monsoon layer can have significant implications for large
       populations in West Africa, in addition to its effects on radiative forcing.

       The significant contribution of long-range emissions towards local pollution in southern West African coastal cities
       highlights the often unique challenges faced in policy creation in developing regions. The population in West Africa
is currently almost 400 million and is expected to more than double in the next 30 years (UN, 2017), with a growing
       proportion living in cities along the southern coast. Thus, the monsoon layer aerosol described in this paper will
       increase the $PM_1$ exposure of a large population by around 8 µg m$^{-3}$ from June to September, based on observational
       evidence presented here. This is a considerable proportion of the 10 µg m$^{-3}$ annual exposure recommended by the
       World Health Organisation (WHO, 2005).

During the dry season (November-January), high concentrations of desert dust from the Sahara and local biomass
       burning are advected into the region. Results presented here show that high levels of particulate are not confined to
       the local dry season but are present throughout much of the year as a result of long-range transport. This regional
       influx of aerosol presents a challenge for future management of air quality in countries across West Africa.

Controlling air quality in these cities cannot be considered solely in terms of reducing local anthropogenic
emissions. Rather, regional- and continental-scale sources of particulate, notably these large biomass burning sources, must be considered. This contrasts with air quality problems encountered in North America and Europe, where urban emissions contribute the majority of air pollution. Solely importing air quality strategies from these regions may therefore be unsuccessful in West Africa, given that transnational transport of particulate plays an important role. Thought should be given to changes in land use practices in countries across the African continent to
reduce the quantity of biomass burning if human exposure to particulate matter is to be limited.

*Author contributions.* SLH, JWT, JB, AMB, SB, JS, CS, CD, TB, RD, AS, DS, CF, JD and IC were involved in the collection, quality assurance and analysis of observational data used in this manuscript. ME, EM, BV and AD developed the model code and carried out simulations. SLH prepared the manuscript with significant contributions from all authors.

*Data availability.* Data from all three aircraft is publicly available on the SEDOO database (baobab.sedoo.fr/DACCIWA).

*Competing interests.* The authors declare that they have no conflict of interest.

*Special issue statement.* This article is part of the special issue *Results of the project "Dynamics-aerosol-chemistry-cloud interactions in West Africa" (DACCIWA).*

*Acknowledgements.* The research leading to these results has received funding from the European Union Seventh Framework Programme (FP7/2007-2013) under grant agreement no. 603502. The lead author was supported by the Natural Environment Research Council Doctoral Training Programme (NERC DTP; grant ref: NE/L002469/1). This paper contains modified Copernicus Atmosphere Monitoring Service Information 2018. The meteorological data used in this study has been provided by the Global Modeling and Assimilation Office (GMAO) at NASA
Goddard Space Flight Center. The participation of AMB, CS, JS and SB on the DLR Falcon 20 in this campaign was made possible by internal funds of the Max Planck Institute for Chemistry in Mainz. We thank the British Antarctic Survey (BAS, operator of the Twin Otter), the Service des Avions Français Instrumentés pour la Recherche en Environnement (SAFIRE, a joint entity of CNRS, Météo-France and CNES and operator of the ATR-42), and the Dutsches Zentrum für Luft-und Raumfahrt (operator of the Falcon 20) for their support during the
aircraft campaign.

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
