# Peer review of "Remote biomass burning dominates southern West African air pollution during the monsoon"

_Atmospheric Chemistry and Physics, 2019_

## Referee Comment (RC1) · Anonymous Referee #2 · 3 May 2019

General remarks.

The main objective of this paper is to present and discuss the airborne aerosol measurements carried out during the DACCIWA campaign in the monsoon layer (ML) of the Gulf of Guinea coast. The objective of the work is very well described and sound because there is still uncertainty about the role of biomass burning (BB) emission in southern and central Africa on the composition of low-level aerosols in the monsoon layer. In addition to the paper Mari et al. (2008), the authors could provide other AMMA publications that raised the issue of the distinction between BB and local pollution emissions by using trace gas or aerosol measurements near the Gulf of Guinea coast during the wet season. The analysis of data from three aircraft under three different chemical regimes (urban outflow, upwind marine and continental background) is

very convincing to support the hypothesis that the background composition of aerosols at the regional level is of critical importance. The analysis of the AMS data indeed demonstrates that the role of BB emission is very likely, but I still believe that other emission sources such as Nigerian off-shore oil fields should also be discussed, given the relatively large amount of sulfate already present in the upwind marine class (>25%) and not in the BBA layer above the monsoon layer (<15%). Section 3.3 on the modelling study showing cloud droplet concentration as a function of aerosol composition is somewhat beyond the scope of this nice paper on airborne data analysis. If the authors want to maintain such a model study, either a thorough discussion on model capabilities to address this issue or at least a comparison between DACCIWA measurements of cloud and aerosol properties and model simulations are necessary.

Detailed remarks and questions:

p.3 line 70 How do the unusual dry conditions modify the paper conclusions about the aerosol composition during the wet season ?

p.5 line 91 Add also a discussion about the characteristics of the AMS instrument since it is a major contribution to the data analysis

p.5 line 97 Comment on the 40% discrepancy between the CPC observations. Typical ?

p. 5 line 107 "surface level aerosol" is misleading. Do you mean aircraft data near the surface or surface network observations ? If surface network observations are used they should be described.

p. 5 line 113 Why is the 1.9 km altitude selected for the ML ? A plot of monsoon layer height distribution as a function of latitude might be useful to justify the altitude level selected, i.e. from aircraft met data.

p. 6 line 128 Are the emissions from the off-shore oil extraction fields included in the EDGAR data ?

[Figure]

p.9 line 197 What is the sulfate and organic aerosol fraction expected for gas flaring emission ? How do you explain the large sulfate contribution for the upwind marine ?

Caption of Fig. 4: Define BBA layer

p.10 Very interesting section on AMS data analysis to demonstrate the major role of BB emissions even within the ML. Are m/z 44 and 28 occurrence characteristic of aging regardless of the aerosol type or specific of the BB aging ?

Caption of Fig. 6 define the dashed line

Fig.6 Might be useful to add "Fresh BB" in the figure within the triangle area

p.13 line 285-290 Do satellite observations (MODIS, MISR, CALIOP) show the extent of the BB plume above the ocean during DACCIWA ?

p.14 line 322 How is the 10% fraction of the BB plume entrained in the ML obtained ?

p.15 I am not convinced that the work with COSMO-ART is a significant added value to this paper as the model performances are not provided to perform such a study on aerosol/cloud interaction in addition to the initial goal of the paper of the role of regional BB within the ML. My feeling is that this section is not necessary and would require a specific publication where the important question aerosol/interaction question is properly introduced with relevant literature and where the model is validated against the DACCIWA data set before being to discuss the low level cloud formation in the ML.

p. 17 line 388-420 This section of the conclusion which advocates for new campaigns could be significantly reduced.

---

## Referee Comment (RC2) · Anonymous Referee #1 · 7 May 2019

This paper discusses airborne aerosol particle measurements over a coastal region from Cote D'Ivoire to Benin. Three extensively-equipped research aircraft are used and data is collected from marine as well as coastal and inland regions during a three-week campaign in 2016.

The focus of the analysis is on chemical composition measured with AMS (on board three aircrafts) and size distribution measured with SMPS (on board one aircraft). The main finding of the paper is that aerosol chemical composition and accumulation mode size distribution within the lowest 2km are very similar over the ocean and inland. This is an important finding as it characterises a significant regional background aerosol load, on top of which urban emissions are emitted. However, the claim that this back-ground aerosol originates from central and southern African biomass burning is not

supported by the data.

Major comments

The average chemical composition observed for <4km aerosol has high SO4 (and NH4) content compared to biomass burning smoke observations - see e.g. Capes et al. (2008) and Vakkari et al. (2014) for observations from African fires. And there is a wealth of observations from e.g. North and South American biomass burning. Actually, the composition of the <4km aerosol is quite close to dry season average composition outside Johannesburg, South Africa (Tiitta et al., 2014). Only the >5km composition looks like pure biomass burning aerosol (OA>80%).

In my opinion the similar chemical composition for urban outflow and upwind marine is an indication of urban emissions being transported over the ocean. A sea breeze circulation can extend up to 150 km from shoreline and could be responsible for such transport, as shown by Flamant et al. (2018) for DACCIWA domain. CALIPSO aerosol profiles might help to show where the observed aerosol layer originates – if it is limited to within 100-200 km of coast or is really connected to the westward transport of central African biomass burning.

Two parameters are missing from the biomass burning aerosol analysis: BC and CO. It would greatly increase the value of this data set if you can provide the PM/CO ratio (and OA/CO) for the different layers, as this value can be (and should be) compared to other biomass burning observations. Likewise, if you have BC (e.g. from aerosol absorption measurements) it would be highly valuable to include it in Fig. 4. BC can easily make 10% of BB aerosol.

Regarding the mass spectra presented in Fig. 5: please apply PMF to the data to check if the same sources contribute as much to the different geographical regions. Please include also >5km BB layer in the analysis.

Regarding COSMO-ART modelling, the only purpose of the model seems to be showing that BB emissions have an effect on cloud droplet size distribution. However, this much is evident from the previous studies cited in the manuscript (see also Rosenfeld et al., 2019). Fig. 8 comparison is relevant only if the model (with BB emissions) gets the aerosol properties correct for the study domain. If you decide to keep the model section, please provide a comparison with observations, and a more detailed description of the model setup. For instance, the model domain (Fig. 2) seems to exclude a substantial fraction of the central African active fires (Fig. 7a).

Minor comments

Line 2. "which releases large concentrations of aerosols into the atmosphere" I think here the amount of aerosol is more relevant than the concentration near source. Same for line 34.

Line 108. Please define the "upwind marine" regime as distance from the shoreline and/or indicate it on the map in Fig. 1.

Line 131. Did you scale the emissions by 3.4 as recommended by Kaiser et al. (2012)?

Line 145. Please define the size range for Aitken and accumulation mode.

Line 164. Are the "N" and "M" for panels b and c correct?

Line 171. Density of 1.3 g/cm3 seems quite low to considering that 40% of the aerosol is inorganic. Please justify the value.

Line 191. Are the CN number concentrations averaged over all CPC's or only for the CPC that had similar cut-off diameter? 14nm cut-off diameter may result in a much lower concentration compared to 3nm cut-off in some cases.

Line 197-201. Please compare with biomass burning plume chemical composition measurements. For regional aerosol Tiitta et al. (2014) measurements in South Africa may be more relevant comparison than global average – please check if there are other more recent observations available.

Line 239-240. "The m/z 44 contribution here correlates well with the total organic mass" Please include correlation coefficient.

Line 279. Should "(a)" and "(b)" be "(b)" and "(c)"?

Line 290-291. I disagree: the large contribution of SO4 suggests significant contribution of non-BB sources.

Line 321-322. "However, results presented here show that in addition, a significant proportion (up to 10%) of the aerosol mass from the biomass burning plume is being entrained into the boundary layer" This, especially the 10%, is not shown here. Please elaborate.

Line 332. The number of CCN is the relevant parameter; aerosol number concentration can be dominated by nucleation mode particles that are too small to act as CCN.

Line 341. Y-label: what is "cloud number concentration"?

Line 407-410. "Previous research has shown that during the monsoon season, submicron particles in southern West Africa absorb moisture and can easily grow to more than double their dry diameter (Deetz et al., 2018; Haslett et al., 2018). This would therefore enhance the aerosol mass loading from these particles, potentially close to the 10 $\mu$g m-3 annual exposure recommended by the World Health Organisation (WHO, 2005)." I think that the air quality limits (and epidemiological studies) are based on dry PM. Please check.

References

Capes, G., Johnson, B., McFiggans, G., Williams, P. I., Haywood, J. and Coe, H.: Aging of biomass burning aerosols over West Africa: Aircraft measurements of chemical composition, microphysical properties, and emission ratios, J. Geophys. Res., 113, doi:10.1029/2008JD009845, 2008.

Flamant, C., Deroubaix, A., Chazette, P., Brito, J., Gaetani, M., Knippertz, P., Fink,

A. H., de Coetlogon, G., Menut, L., Colomb, A., Denjean, C., Meynadier, R., Rosen-berg, P., Dupuy, R., Dominutti, P., Duplissy, J., Bourrianne, T., Schwarzenboeck, A., Ramonet, M. and Totems, J.: Aerosol distribution in the northern Gulf of Guinea: local anthropogenic sources, long-range transport, and the role of coastal shallow circulations, Atmos. Chem. Phys., 18, 12363–12389, doi:10.5194/acp-18-12363-2018, 2018.

Kaiser, J. W., Heil, A., Andreae, M. O., Benedetti, A., Chubarova, N., Jones, L., Mor-crette, J.-J., Razinger, M., Schultz, M. G., Suttie, M. and van der Werf, G. R.: Biomass burning emissions estimated with a global fire assimilation system based on observed fire radiative power, Biogeosciences, 9, 527–554, doi:10.5194/bg-9-527-2012, 2012.

Rosenfeld, D., Zhu, Y., Wang, M., Zheng, Y., Goren, T. and Yu, S.: Aerosol-driven droplet concentrations dominate coverage and water of oceanic low-level clouds, Science, 363, eaav0566, doi:10.1126/science.aav0566, 2019.

Tiitta, P., Vakkari, V., Croteau, P., Beukes, J. P., van Zyl, P. G., Josipovic, M., Venter, A. D., Jaars, K., Pienaar, J. J., Ng, N. L., Canagaratna, M. R., Jayne, J. T., Kermi-nen, V.-M., Kokkola, H., Kulmala, M., Laaksonen, A., Worsnop, D. R. and Laakso, L.: Chemical composition, main sources and temporal variability of PM1 aerosols in southern African grassland, Atmos. Chem. Phys., 14, 1909–1927, doi:10.5194/acp-14-1909-2014, 2014.

Vakkari, V., Kerminen, V.-M., Beukes, J. P., Tiitta, P., van Zyl, P. G., Josipovic, M., Venter, A. D., Jaars, K., Worsnop, D. R., Kulmala, M. and Laakso, L.: Rapid changes in biomass burning aerosols by atmospheric oxidation, Geophys. Res. Lett., 41, 2644–2651, doi:10.1002/2014GL059396, 2014.
* * *

---

## Author Comment (AC1) · 5 Aug 2019

We thank the reviewer for their comments and suggestions. Please find our responses attached. The manuscript is included, with changes in red.
* * *

---

## Author Comment (AC2) · 5 Aug 2019

We thank the reviewers for taking the time to assess this manuscript and for their insights and suggestions for its improvement. Based on these comments, we have made revisions to our original manuscript and provided responses individually to the suggestions below. The reviewer's comments are in black and our responses are in red.

General remarks.

The main objective of this paper is to present and discuss the airborne aerosol measurements carried out during the DACCIWA campaign in the monsoon layer (ML) of the Gulf of Guinea coast. The objective of the work is very well described and sound because there is still uncertainty about the role of biomass burning (BB) emission in southern and central Africa on the composition of low-level aerosols in the monsoon layer. In addition to the paper Mari et al. (2008), the authors could provide other AMMA publications that raised the issue of the distinction between BB and local pollution emissions by using trace gas or aerosol measurements near the Gulf of Guinea coast during the wet season. The analysis of data from three aircraft under three different chemical regimes (urban outflow, upwind marine and continental background) is very convincing to support the hypothesis that the background composition of aerosols at the regional level is of critical importance. The analysis of the AMS data indeed demonstrates that the role of BB emission is very likely, but I still believe that other emission sources such as Nigerian off-shore oil fields should also be discussed, given the relatively large amount of sulfate already present in the upwind marine class (>25%) and not in the BBA layer above the monsoon layer (<15%). Section 3.3 on the modelling study showing cloud droplet concentration as a function of aerosol composition is somewhat beyond the scope of this nice paper on airborne data analysis. If the authors want to maintain such a model study, either a thorough discussion on model capabilities to address this issue or at least a comparison between DACCIWA measurements of cloud and aerosol properties and model simulations are necessary.

We have taken these comments into consideration and integrated suggestions into the manuscript. More references related to AMMA have been included in discussing the presence of a biomass burning layer between 2 and 4 km (Murphy et al., 2010; Reeves et al., 2010; Sauvage et al., 2005; lines 37-38). In particular, a modelling study based on AMMA data by Deroubaix et al. (2018) has now been discussed in more detail in lines 50-54 and lines 452-456. This study assessed the relative contributions of biomass burning aerosol from central Africa and local anthropogenic pollution in southern West Africa during the AMMA campaign. Similar to our results, this study showed that long-range transport of biomass burning aerosol is likely to have a significant influence on pollution in the region (contributing around 52% below 1 km).

We have included some discussion of off-shore oil fields into the text in lines 405-408. Emissions from these oil fields were observed during the DACCIWA campaign, with several flights by the Falcon being carried out specifically to investigate their impact. The emissions were found to have very characteristic profiles, with narrow plumes of increased NOx, CO and PM being observed close to the sources and

dissipating after around 40 km (Brocchi et al., 2019). Oil field plumes were not observed during the flights included in this analysis, however. The primary particulate emission from oil flaring is black carbon (Fortner et al., 2012). It is likely that the sulfate observed over the ocean in our case was from a mixture of sea salt and DMS (dimethyl sulphate) emissions, which is typical in marine air masses (Choi et al., 2017; Fitzgerald, 1991).

We have worked to integrate our modelling results more fully into the manuscript, as we feel this both provides a further body of evidence to support our conclusions, and that it was a valuable addition to the manuscript to consider the impact of this aerosol mass on cloud properties. A second model, GEOS-Chem, was also run as part of this project and we have now included these results as well as those from COSMO-ART. We have shown these model results earlier in the paper alongside observational results and have discussed more fully the ways in which they support our conclusions.

Detailed remarks and questions:

p.3 line 70 How do the unusual dry conditions modify the paper conclusions about the aerosol composition during the wet season ?

Our measurements took place during June-July, when the bulk of the monsoon precipitation is typically far north of the coast, over the Sahel. In southern West Africa this is often called the little dry season, as there is only occasional rain. Although it was slightly drier than usual during 2016 due to the northward-shifted intertropical discontinuity, the overall conditions were typical of this time of year in being dominated by southerlies. Thus, aerosol composition was controlled by long-range transport and local sources, with very little dust, which is the normal situation. We have reworded this phrase in the manuscript to acknowledge that the differences from normal conditions were only minor, in lines 82-84.

p.5 line 91 Add also a discussion about the characteristics of the AMS instrument since it is a major contribution to the data analysis

More detail has been added to the description of the AMS is in lines 93-98.

p.5 line 97 Comment on the 40% discrepancy between the CPC observations. Typical ?

This was an oversight – there was a 40% discrepancy over the whole campaign, largely due to sampling bias in the locations in which the aircraft flew. When comparing results from when aircraft were flying in comparable locations, however, this discrepancy was much lower. Along the Lomé-Savè transects described in the text that were used to compare the instruments, there was only a 10% difference between the CPCs on different aircraft. This has been updated in the text in line 115. More details from this comparison are shown here in Fig. 1.

It was not possible during the campaign to compare the different instruments directly. Therefore, there may still be sampling bias in these results, which may explain the difference in spread between the Twin Otter CPC and the Falcon CPC.

[Figure]

**Figure 1: A comparison of the frequency distribution of CPC measurements in the continental background regime for the three different aircraft. This shows close agreement between the median CPC counts for each of the aircraft.**

p. 5 line 107 "surface level aerosol" is misleading. Do you mean aircraft data near the surface or surface network observations ? If surface network observations are used they should be described.

This term has now been removed from this description. We were referring to aerosol in the monsoon layer.

p. 5 line 113 Why is the 1.9 km altitude selected for the ML ? A plot of monsoon layer height distribution as a function of latitude might be useful to justify the altitude level selected, i.e. from aircraft met data.

In this paper we are using the height of the monsoon layer, as described by Kalthoff et al. (2018), which was established to be 1.9 km on average over Savè, which is about 180km north of the coast in Benin. As the monsoon layer typically gets shallower towards the north, we assume this value to be reasonable for our purposes. This is the height of the deep, moist layer that transports moisture into West Africa from the southwest. Long-range transport in the monsoon layer controls the aerosol concentration in the boundary layer. We have updated the terminology in the manuscript to reflect this distinction more accurately and included a comparison between the two terms in lines 131-133 for clarity.

p. 6 line 128 Are the emissions from the off-shore oil extraction fields included in the EDGAR data ?

Emissions from the off-shore oil extraction fields are included in the EDGAR data. There may be some double counting of these emissions; however, this effect is likely small. The EDGAR HTAP_v2 flaring approach neglects all offshore flaring and was done based on rural population proxy data in 2010. The only double count that could be considerable is in the Niger Delta (Nigeria) and it is unlikely that plumes from here would be observed in the DACCIWA region, based on average wind directions.

p.9 line 197 What is the sulfate and organic aerosol fraction expected for gas flaring emission ?

The overwhelming majority of aerosol emitted from gas flaring is released as black carbon (Fortner et al., 2012). Studies from DACCIWA have shown that over our campaign region, gas flaring emissions were highly recognisable as narrow plumes of pollution that were strongest close to the source (Brocchi et al., 2019). These characteristic spikes in CO, NOx and aerosol concentrations were not visible during the flights analysed here, making it unlikely that gas flaring is a significant contribution to the aerosol

background we observed. An acknowledgement of these emissions has now been added to the text in lines 405-408.

How do you explain the large sulfate contribution for the upwind marine ?

Sulfate is typically found over the ocean, from both sea salt and the oxidation products of DMS. Around 2 µg m$^{-3}$ from these sources combined is to be expected. This explanation has been added to the text in lines 274-278 (Choi et al., 2017; Fitzgerald, 1991).

Caption of Fig. 4: Define BBA layer

This has been added to the text.

p.10 Very interesting section on AMS data analysis to demonstrate the major role of BB emissions even within the ML. Are m/z 44 and 28 occurrence characteristic of aging regardless of the aerosol type or specific of the BB aging ?

M/z 44 and 28 are characteristic of general aged organic aerosol – this is not specific to biomass burning. We have written this now as "aged organic aerosol" instead of just "aged aerosol" in line 352, to underline that this is not specific.

Caption of Fig. 6 define the dashed line

This has now been defined.

Fig.6 Might be useful to add "Fresh BB" in the figure within the triangle area

This has been added to the figure.

p.13 line 285-290 Do satellite observations (MODIS, MISR, CALIOP) show the extent of the BB plume above the ocean during DACCIWA ?

It is not possible to isolate the extent of the BB layer in the monsoon layer using satellite data, due to the semi-permanent clouds over the Atlantic and the much larger BB plume that is often found at higher elevation, between 2 and 4 km. Both of these can obscure the plume at lower levels, making it difficult to ascertain its extent. CALIPSO data, which uses a lidar to produce vertical cross sections, indicates that there is an unbroken layer of aerosol below the clouds stretching from the Atlantic Ocean to the east of central Africa up to the DACCIWA region. This is illustrated with the below plot (Fig. 2) from 14[th] July, one of the days when the ATR was flying over the ocean south of the coast.

[Figure]

**Fig. 2: Results from the CALIPSO satellite showing aerosol present in the boundary layer below the clouds over the Atlantic Ocean.**

p.14 line 322 How is the 10% fraction of the BB plume entrained in the ML obtained ?

This fraction was based on the difference in concentration between the monsoon layer aerosol and the elevated biomass burning aerosol. However, after further discussion, we have removed this number as it was not robustly supported.

p.15 I am not convinced that the work with COSMO-ART is a significant added value to this paper as the model performances are not provided to perform such a study on aerosol/cloud interaction in addition to the initial goal of the paper of the role of regional BB within the ML. My feeling is that this section is not necessary and would require a specific publication where the important question aerosol/interaction question is properly introduced with relevant literature and where the model is validated against the DACCIWA data set before being to discuss the low level cloud formation in the ML.

We agree that showing only the modelled impact of aerosol changes on clouds created a disconnect from the rest of the paper. Nevertheless, we believe that the model has interesting information to offer to this study that makes our argumentation stronger. Therefore, we have decided to extend this section to link it better to the rest, as well as including additional results from the GEOS-Chem model, described in lines 176-193. This allows us to explore simulated averages over the entire campaign, as well as a simulation of a single day. In the revised version, we now include a comparison of the simulated with the observed mass concentration to show that the model produces realistic fields, particularly over the upwind marine area. We show that switching off biomass burning in both models results in a decrease in aerosol concentration over the upwind marine area of around 75% (lines 253-257), which supports our interpretation of long-range transport from central Africa. After this additional discussion of modelling results, we finally show the impact on clouds, as a result that we cannot retrieve from the observations alone. We modified the text to justify the use of a model to complement our results much better than in the original version.

p. 17 line 388-420 This section of the conclusion which advocates for new campaigns could be significantly reduced.

This section has now been reduced.

Brocchi, V., Krysztofiak, G., Deroubaix, A., Stratmann, G., Sauer, D., Schlager, H., Deetz, K., Dayma, G., Robert, C., Chevriere, S. and Catoire, V.: Local air pollution from oil rig emissions observed during the airborne DACCIWA campaign, Atmos. Chem. Phys. Discuss., doi: 10.5194/acp-2019-27, 2019.

Choi, Y., Rhee, T. S., Collette Jr. J. L., Taehyun, P., Park, S.-M., Seo, B.-K., Park, G., Park, K., Lee, T.: Aerosol concentrations and composition in the North Pacific marine boundary layer, Atmos. Environ., 171, 165-172, doi: 10.1016/j.atmosenv.2017.09.047, 2017.

Deroubaix, A., Flamant, C., Menut, L., Siour, G., Mailler, S., Turquety, S., Briant, R., Khvorostyanov, D. and Crumeyrolle, S.: Interactions of atmospheric gases and aerosols with the monsoon dynamics over the Sudano-Guinean region during AMMA, Atmos. Chem. Phys., 18, 445-465, doi: 10.5194/acp-18-445-2018, 2018.

Fitzgerald, J. W.: Marine aerosols: a review, Atmos. Environ., 25A, 3/4, 533-545, doi: 10.1016/0960-1686(91)90050-H, 1991.

Fortner, E. C., Brooks, W. A., Onasch, T. B., Canagaratna, M. R., Massolie, P., Jayne, J. T., Franklin, J. P., Knighton, W. B., Wormhoudt, J., Worsnop, D. R., Kolb, C. E. and Herndon, S. C.: Particulate emissions measured during the TCEQ comprehensive flare emission study, Ind. Eng. Chem. Rs., 51, 12586-12592, doi: 10.1021/ie202692y, 2012.

[revised manuscript text omitted]

---

## Author Comment (AC3) · 5 Aug 2019

We thank the reviewers for taking the time to assess this manuscript and for their insights and suggestions for its improvement. Based on these comments, we have made revisions to our original manuscript and provided responses individually to the suggestions, below. The reviewer's comments are in black and our responses are in red.

This paper discusses airborne aerosol particle measurements over a coastal region from Cote D'Ivoire to Benin. Three extensively-equipped research aircraft are used and data is collected from marine as well as coastal and inland regions during a threeweek campaign in 2016. The focus of the analysis is on chemical composition measured with AMS (on board three aircrafts) and size distribution measured with SMPS (on board one aircraft). The main finding of the paper is that aerosol chemical composition and accumulation mode size distribution within the lowest 2km are very similar over the ocean and inland. This is an important finding as it characterises a significant regional background aerosol load, on top of which urban emissions are emitted. However, the claim that this background aerosol originates from central and southern African biomass burning is not supported by the data.

We have taken significant steps in this revised version of the manuscript to address the reviewer's concern here that the data do not fully support our conclusions. We have extended our descriptions of the meteorological conditions in the region and included a discussion about the extent of the sea-breeze circulation to the south of the West African coast. We have included further exploration of the chemistry in the manuscript, including an extended discussion of the OA/CO ratio. In addition, we have worked to integrate our modelling results more fully with the rest of the paper. A second model, GEOS-Chem, was also run as part of this project and we have now included these results as well as those from COSMO-ART. We have shown these model results earlier in the paper alongside observational results and discussed more fully the ways in which they support our conclusions. We believe these revisions have strengthened the evidence supporting our conclusions.

Major comments

The average chemical composition observed for <4km aerosol has high SO4 (and NH4) content compared to biomass burning smoke observations - see e.g. Capes et al. (2008) and Vakkari et al. (2014) for observations from African fires. And there is a wealth of observations from e.g. North and South American biomass burning. Actually, the composition of the <4km aerosol is quite close to dry season average composition outside Johannesburg, South Africa (Tiitta et al., 2014). Only the >5km composition looks like pure biomass burning aerosol (OA>80%).

A comparison to these studies has now been included in lines 279-284.

It is likely that the incoming aerosol in the monsoon layer has also been influenced by marine aerosol, which explains the high sulfate concentration. This was not properly discussed in the text before, but has now been added to lines 274-278.

During the DACCIWA campaign the air above 5 km mostly came from an easterly-northeasterly direction, roughly from the direction of Sudan and Chad (see, for example, the back trajectories shown below). Therefore, this air mass is likely to be primarily influenced by desert dust (which cannot be measured by conventional AMS techniques). Non-refractory aerosol concentrations were low (~0.6 µg m$^{-3}$ organics) and analysis of the AMS data indicates that at least some of the organics at this level are formed in-situ. Back trajectories at this level do not cross the burning area in central Africa. It is therefore unlikely that aerosol at this level originates purely from biomass burning. The sources influencing aerosol composition at this altitude are very different to those influencing the lower layers.

The presence of a BBA layer originating from central/southern African fires between 2 and 4 km is well-established (Chatfield et al., 1998; Das et al., 2017; Mari et al., 2008; Murphy et al., 2010; Sauvage et al., 2005). The similarities between aerosol properties in this layer and in the monsoon layer over the sea are a strong indication that there is a common source.

[Figure]

**Figure 1: Back trajectories showing the origin of the air mass in the upper troposphere (> 5 km).**

In my opinion the similar chemical composition for urban outflow and upwind marine is an indication of urban emissions being transported over the ocean. A sea breeze circulation can extend up to 150 km from shoreline and could be responsible for such transport, as shown by Flamant et al. (2018) for DACCIWA domain. CALIPSO aerosol profiles might help to show where the observed aerosol layer originates – if it is limited to within 100-200 km of coast or is really connected to the westward transport of central African biomass burning.

It is true that sea and land breeze circulations can reach the spatial extent you are mentioning, but this is not the case in southern West Africa during summer. Here, the sea-breeze circulation superposes with the strong southerly monsoon flow, which largely impedes the transport from land to sea. A pilot balloon climatology by Guedje et al. (2019) for Cotonou reveals that during July–September the flow below 1km is exclusively from southerly quadrants, while above that a weak northerly component does occur occasionally. As shown by Flamant et al. (2018)'s Fig. 8 derived from numerical tracer experiments for a case study, city pollution is mostly transported inland and hardly reaches beyond 50 km south of the shoreline and even less near the surface where the southerlies are strongest. Radiosonde measurements from coastal stations shown in Fig. 4 of Flamant et al. show that there is hardly a northerly component in the wind at all. For the upwind marine domain, we discarded data less than 20

km south of the coast, thus largely filtering out a potential influence from West African cities. The analysis by Flamant et al. (2018) focused on one ATR flight, which took place on 2 July 2016 and was chosen specifically due to its proximity to the coast. Data from this flight were not included in our analysis here, as it was not considered to represent the marine background.

The southerlies in the monsoon layer are very stable during the summer, with an onshore flow during both the day and the night (Flamant et al., 2018; Guedje et al., 2019), leaving little opportunity for a pollution transport far to the south of the coast. This has now been explained in more detail in the text in lines 396-405.

The semi-permanent cloud cover over the Atlantic and adjacent land makes it very difficult to determine the presence of a biomass burning layer from CALIPSO to the south of the coast, though there are indications that it is present. See, for example, the figure shown below, which is from CALIPSO on 14th July 2016, one of the days that the ATR aircraft flew over the Atlantic.

[Figure]

**Fig. 2: Results from the CALIPSO satellite showing aerosol present in the boundary layer below the clouds over the Atlantic Ocean.**

Two parameters are missing from the biomass burning aerosol analysis: BC and CO. It would greatly increase the value of this data set if you can provide the PM/CO ratio (and OA/CO) for the different layers, as this value can be (and should be) compared to other biomass burning observations. Likewise, if you have BC (e.g. from aerosol absorption measurements) it would be highly valuable to include it in Fig. 4. BC can easily make 10% of BB aerosol.

An analysis of OA/CO in the different regimes has been included in lines 353-381, alongside a figure showing this relationship. These have been compared to other values in the literature. We have also now shown in Fig. 7 that our model was unable to reproduce the observed CO concentrations over the ocean without the includion of biomass burning in central/southern Africa.

The sparse coverage of BC measurements during this campaign and significant disagreements between instruments on different aircraft made it very difficult to establish BC concentrations in each regime. We are therefore, unfortunately, unable to include BC observations here.

Regarding the mass spectra presented in Fig. 5: please apply PMF to the data to check if the same sources contribute as much to the different geographical regions. Please include also >5km BB layer in the analysis.

PMF has already been carried out for this dataset and the results are available in Brito et al. (2018). Three factors were found: fresh urban aerosol, which included tracers of traffic emission and fresh biomass burning, was found primarily close to the cities; aged urban aerosol, which was mostly present in plumes downwind of large cities such as Lomé or Abidjan; and oxidised organic aerosol, which was found to be present across the whole region, including over the ocean. This third factor corresponds to the background of aged organic aerosol that we discuss here, and the mass spectrum of this factor is almost identical to those shown here for the elevated biomass burning layer and for the upwind marine aerosol. A reference to this study has been included in lines 309-312.

Regarding COSMO-ART modelling, the only purpose of the model seems to be show ing that BB emissions have an effect on cloud droplet size distribution. However, this much is evident from the previous studies cited in the manuscript (see also Rosenfeld et al., 2019).

Indeed there were a lot of studies in the past that assessed the interaction of biomass burning aerosols with cloud formation (the paper of Rosenfeld et al. was published after our submission). Nevertheless, the results are still subject to large uncertainties due to the methods applied in the individual studies and justify addressing the topic with an alternative method. Here, we use a fully interactive aerosol-chemistry scheme, together with a detailed cloud microphysics scheme instead of prescribing the aerosol properties, as it was often done in the past.

Fig. 8 comparison is relevant only if the model (with BB emissions) gets the aerosol properties correct for the study domain. If you decide to keep the model section, please provide a comparison with observations, and a more detailed description of the model setup.

We extended the model description and added a comparison with observations.

For instance, the model domain (Fig. 2) seems to exclude a substantial fraction of the central African active fires (Fig. 7a).

As we are using MOZART results as boundary conditions for our model results we do not expect that we are losing substantial contributions of biomass burning over Central Africa. We added this information to the model description.

Minor comments

Line 2. "which releases large concentrations of aerosols into the atmosphere" I think here the amount of aerosol is more relevant than the concentration near source. Same for line 34.

This has been updated to "large quantities" in the manuscript, in both cases.

Line 108. Please define the "upwind marine" regime as distance from the shoreline and/or indicate it on the map in Fig. 1.

This has been updated in line 127: we are only taking into account data more than 20 km south of the shoreline.

Line 131. Did you scale the emissions by 3.4 as recommended by Kaiser et al. (2012)?

We made no correction as we are not using the GFAS emission data. We have our own emission scheme described in Walter et al. (2016). The only thing we are taking from GFAS is the fire radiative power. Therefore we do not need to scale our emissions.

Line 145. Please define the size range for Aitken and accumulation mode.

This has now been defined as particles smaller than 100 nm.

Line 164. Are the "N" and "M" for panels b and c correct?

These were the wrong way round – they have now been switched back to the correct positions.

Line 171. Density of 1.3 g/cm3 seems quite low to considering that 40% of the aerosol is inorganic. Please justify the value.

Yes this was an old value and is too low. This has been changed to a new value of 1.6 g cm$^{-3}$, which was calculated by Haslett et al. (2019). These values are now closer to the measured aerosol mass concentrations. We thank the reviewer for spotting this.

Line 191. Are the CN number concentrations averaged over all CPC's or only for the CPC that had similar cut-off diameter? 14nm cut-off diameter may result in a much lower concentration compared to 3nm cut-off in some cases.

The CN concentrations were averaged across all CPCs. There was found to be very little discrepancy between the CPCs within similar air masses (<10%), which suggests that the contribution of particles between 3 and 14 nm was minimal during this campaign. In addition, there are likely to be inlet and tubing line effects leading to substantial particle losses in the sub-10 nm size range. Therefore, the real system cutoffs are likely not as different as the CPC technical specifications might suggest.

Line 197-201. Please compare with biomass burning plume chemical composition measurements. For regional aerosol Tiitta et al. (2014) measurements in South Africa may be more relevant comparison than global average – please check if there are other more recent observations available.

A more in-depth comparison to other measurements has now been added to lines 279-284.

Line 239-240. "The m/z 44 contribution here correlates well with the total organic mass" Please include correlation coefficient.

The correlation coefficient has been added to the text in line 326.

Line 279. Should "(a)" and "(b)" be "(b)" and "(c)"?

This has been updated.

Line 290-291. I disagree: the large contribution of SO4 suggests significant contribution of non-BB sources.

Sulfate is typically found over the ocean, from both sea salt and the oxidation products of DMS. Around 2 µg m$^{-3}$ from these sources combined is to be expected, which is around what we see here. This explanation has been added to the text in lines 274-278 (Choi et al., 2017; Fitzgerald, 1991).

Line 321-322. "However, results presented here show that in addition, a significant proportion (up to 10%) of the aerosol mass from the biomass burning plume is being entrained into the boundary layer" This, especially the 10%, is not shown here. Please elaborate.

This fraction was based on the difference in concentration between the monsoon layer aerosol and the elevated biomass burning aerosol. However, after further discussion, we have removed this number as it was not robustly backed up.

Line 332. The number of CCN is the relevant parameter; aerosol number concentration can be dominated by nucleation mode particles that are too small to act as CCN.

This has been updated in the text.

Line 341. Y-label: what is "cloud number concentration"?

Cloud droplet number concentration. This has been updated.

Line 407-410. "Previous research has shown that during the monsoon season, submicron particles in southern West Africa absorb moisture and can easily grow to more than double their dry diameter (Deetz et al., 2018; Haslett et al., 2018). This would therefore enhance the aerosol mass loading from these particles, potentially close to the 10 µg m-3 annual exposure recommended by the World Health Organisation (WHO, 2005)." I think that the air quality limits (and epidemiological studies) are based on dry PM. Please check.

The text has been altered to take this into account.

[revised manuscript text omitted]